



# Screening of cloud microorganisms isolated at the puy de Dôme (France) station for the production of biosurfactants

Pascal Renard[1,2], Isabelle Canet[1], Martine Sancelme[1], Nolwenn Wirgot[1,2], Laurent Deguillaume[3], Anne-Marie Delort[1,2]

[1] Institut de Chimie de Clermont-Ferrand, Université Clermont Auvergne, Université Blaise Pascal, BP 10448, F-63000 Clermont-Ferrand, France

[2] CNRS, UMR 6296, ICCF, F-63171 Aubière, France

[3] Laboratoire de Météorologie Physique/OPGC, Université Clermont Auvergne, Université Blaise Pascal, BP 10448, F-63000 Clermont-Ferrand, France

*Correspondence to*: Anne-Marie Delort (A-Marie.Delort@univ-bpclermont.fr)

**Abstract.** 480 microorganisms collected from 39 clouds sampled at the puy de Dôme station (alt. 1465 m, 45°46'19" N, 2°57'52" E, Massif Central, France) were isolated and identified. This unique collection was screened for biosurfactant (surfactants of microbial origin) production by measuring the surface tension (σ) of crude extracts, consisting of the supernatants of the pure cultures by the pendant drop technique. 41% of the tested strains are active producers (σ < 55mN m$^{-1}$), 7% being extremely active (σ < 30 mN m$^{-1}$). The most efficient biosurfactant producers (σ < 45 mN m$^{-1}$) belong to few bacterial genera (Pseudomonas and Xanthomonas) from the ϒ-Proteobacteria class (78%), and yeast genus (Udeniomyces) from the Basidiomycota phylum (11%). Some Bacillus strains from the Firmicutes phylum are also active but represent a very small fraction of the collected population. Strains from the Actinobacteria phylum are mainly present in our collection with a moderate biosurfactant production (45 < σ < 55 mN m$^{-1}$). Pseudomonas (ϒ-Proteobacteria) which is the most frequently found genus in clouds and whose some species are issued from the phyllosphere is the dominant group for the production of biosurfactants. Statistical analyses showed some positive correlations between the origin of air masses and chemical composition of cloud waters with the presence of biosurfactant-producing microorganisms, suggesting a "biogeography" of this production. Finally we discuss the potential impact of the production of biosurfactants by cloud microorganisms on atmospheric processes and human health.

**Key words:** Biosurfactants, cloud condensation nuclei, microorganisms

## 1 Introduction

Atmospheric aerosol particles act as cloud condensation nuclei (CCN) upon which liquid droplets can form. More aerosols result in a larger concentration of smaller droplets, leading to a brighter cloud (Twomey effect). However, owing to their complexity, all aerosol–cloud–interactions (ACI) can amplify or dampen this effect. Therefore, ACI, in particular CCN activation, still account for major uncertainties in predictions of global climate and future climate change (Boucher et al., 2013).





Among organic aerosol, water soluble organic compounds (WSOC) represent a significant fraction of tropospheric aerosol mass (Kanakidou et al., 2005; Murphy et al., 2006; Saxena and Hildemann, 1996; Zenchelsky and Youssefi, 1979; Zhang et al., 2007) and constitute a complex mixture of neutral and acidic polar organic compounds (Decesari et al., 2001). As some WSOC are amphipathic compounds, they can therefore act as surfactants (surface-active agents), by partitioning between the droplet gas-liquid interface and the bulk volume. Seidl and Hänel (1983) were among the first to estimate concentrations of surface-active soluble substances on rainwater and wet aerosol by measuring the lowering of the surface tension. Capel et al. (1990) correlated the surface tension of fog samples with their dissolved organic carbon content. Hitzenberger et al. (2002) observed slight surface tension decreases for most of the 23 cloud water samples collected in mountainous and sparsely populated area.

Some groups (Decesari et al., 2005; Facchini et al., 1999) renewed research on surface-active compounds in the atmosphere with a great deal of speculation about their potential impact on climate (Brimblecombe and Latif, 2004). Indeed, surfactants can affect cloud droplet growth in two main ways described by Köhler equation (Köhler, 1936): by increasing soluble mass ("Raoult term") and by decreasing cloud droplet surface tension ("Kelvin term") (Decesari et al., 2003; Facchini et al., 1999; Lance et al., 2004; Mircea et al., 2002; Rodhe, 1999; Shulman et al., 1996). Thus, taking into account both the solute concentration increase and the surface tension decrease, Mircea et al. (2002) calculated a substantial attenuation of the aerosol critical supersaturation, which resulted in a significant increase in CCN number concentration. By adding surfactant to the gas-aerosol interface, Sareen et al. (2013) assessed significant enhancements in CCN activity, up to 7.5% reduction in critical dry diameter for activation, which for ambient aerosol would lead to a 10% increase in the cloud droplet number concentration. Facchini et al. (1999) estimated such a population rise, due to surfactant, might result, in all stratus clouds, in an 1% increase in albedo, and subsequently in a calculated global radiative forcing of -1 W m$^{-2}$.

In addition to surfactant impact on CCN activity *via* both Raoult and Kelvin terms, a third effect has to be taken in account. Surfactants can either enhance or slow down the transfer of water across surface, according to the hydrophilic or hydrophobic nature of this aerosol organic coating (Aumann and Tabazadeh, 2008; Chakraborty and Zachariah, 2011; Feingold and Chuang, 2002; Rudich, 2003). These organic coats are common on aerosol particles and might retard evaporation of molecules present in the water phase, reduce gas transfer, influence chemical reactions, and alter absorption or reflection properties of aerosols (Clifford et al., 2007; Decesari et al., 2003; Gill et al., 1983; Gilman and Vaida, 2006). Nenes et al. (2002) studied the sensitivity of cloud droplet number concentration to different chemical factors as dissolution of soluble gases and solutes or formation of organic films at the droplet surface, and demonstrated that these chemical effects on droplet activation could be significant. In the same way, the $^•$OH heterogeneous reactions that can occur on organics in the troposphere can significantly modify the hygroscopic properties and CCN ability of these organic surfaces (Bertram et al., 2001; Ellison et al., 1999), and thus may play an important role in the Earth's radiative balance by affecting the properties of clouds, *e.g.*, Twomey effect and cloud life time (Aumann et al., 2010; Nenes et al., 2002; Rodhe, 1999).





Recent experiments were devoted to CCN enhancement resulting from biogenic influence (Facchini et al., 2000; O'Dowd et al., 2002; Svenningsson et al., 2006). Thus, some of the atmospheric organic compounds, such as pinic and pinonic acids (produced by oxidation of terpenes in organic vapors released from the canopy) were described as able to depress the surface tension of water even at very low concentrations (Li et al., 2010; O'Dowd et al., 2002). Another important class of hydrophobic WSOC is humic-like substances (HULIS). These complex mixtures of high molecular weight compounds can depress surface tension in fog water samples by 15-20% (Decesari et al., 2003; Dinar et al., 2006; Facchini et al., 2000).

Nevertheless, if number of organic surface-active compounds have been detected in aerosol particles and cloud droplets, a large fraction of WSOC is still poorly characterized (Herckes et al., 2013). Moreover, due to their limited concentrations in aerosols, the question remains if the atmospheric organic surfactants have to be considered as able to decrease the surface tension of atmospheric water, or to contribute to CCN properties of atmospheric particles (McFiggans et al., 2006).

In the last decade, few investigators identified strong organic surfactants in atmospheric aerosols (Baduel et al., 2012). Their exceptional tension-active properties suggested that they could be of microbial origin, "biosurfactants", and could affect cloud formation (Ekström et al., 2010; Nozière et al., 2014). Biosurfactants are secondary metabolites produced by microorganisms including low molecular biosurfactants (mainly glycolipids and lipopeptides) but also high molecular mass biosurfactants (polysaccharides, proteins, lipopolysaccharides, lipoproteins or complex mixtures of these polymers) (Gautam and Tyagi, 2006; Rosenberg and Ron, 1999). These amphiphilic compounds were able to lower the surface tension of atmospheric water below 30 mN m$^{-1}$ (*i.e.*, > -40% as compared to pure water) and for concentrations 5 or 6 orders of magnitude lower than organic acids (Ekström et al., 2010). By comparison, HULIS depressed surface tension in fog water samples by 20% at a 100 mgC L$^{-1}$ (Facchini et al., 2000).

Baduel et al. (2012) measured such surface tensions in summer samples, which would be consistent with the high biogenic activity during that season. Ahern et al. (2007) have shown that, from microorganisms isolated from clouds and rain waters, fluorescent pseudomonads isolates were able to produce biosurfactants. This was the first and sole report on exploration of the potential production of biosurfactants by microorganisms isolated from the clouds environment. Biosurfactants could be both directly issued from the Earth's surface during aerosolization processes or be produced directly in cloud waters. However, a multitude of bacteria, fungi and yeasts have been shown to be metabolically active in clouds (Amato et al., 2005, 2007a; Hill et al., 2007; Sattler et al., 2001; Vaïtilingom et al., 2012, 2013). These microorganisms are able to survive and resist to atmospheric stresses (Delort et al., 2010; Joly et al., 2015).

Poorly considered in the atmosphere, biosurfactants were more extensively studied in soil- and plant-associated environments. Indeed, biosurfactants-producing bacteria are well-known in both undisturbed and contaminated soils (Bodour et al., 2003; Raaijmakers et al., 2010). Biosurfactants are for instance investigated for their capacity to remove heavy metals, and bioremediation is one of the main industrial applications of biosurfactants (Banat et al., 2010; Mulligan, 2009). Finally, in terms of microbial life and activity, biosurfactants play a key role in bacterial cell motility, solubilization of organic compound, formation and disruption of microbial biofilms or



anti-microbial activity (Chrzanowski et al., 2012; D'aes et al., 2010; Mann and Wozniak, 2012; Raaijmakers et al., 2010; Ron and Rosenberg, 2001).

Within a project devoted to atmospheric surfactants and to the study of their effect on cloud droplet formation, we focus on biosurfactant-producing microorganisms present in the atmospheric waters. Cloud water samples are collected at the puy de Dôme station (France) that belongs to the GAW (Global Atmosphere Watch) stations network. 480 bacterial and yeast strains are isolated and identified. This unique collection of microorganisms is screened to identify biosurfactant-producing microorganisms. Surface tension of crude extracts, consisting of the supernatants of the pure culture, is determined by the pendant drop technique (Hansen and Rødsrud, 1991). In order to evaluate the potential correlation between the origin of air masses and composition of cloud waters and the presence of biosurfactant-producing microorganisms, statistical analyses are performed. Finally we discuss the potential impact of the production of biosurfactants by cloud microorganisms on atmospheric physicochemical processes.

## 2 Materials and Methods

### 2.1 Cloud sampling and physicochemical characterization of cloud water samples

Cloud water samples are collected thanks to cloud droplet impactor sterilized by autoclave and installed on the summit of the puy de Dôme mountain (1465 m above sea level, 45°46'19"N, 2°57'52"E, Massif Central). Collected clouds are non-precipitating and non-convective. Hereafter, we work on 480 microbial strains collected during 39 clouds events, from 2004 to 2014. The physicochemical content of the aqueous cloud samples is characterized (concentrations of organic acids, inorganic ions and pH, see Table S1 in the supplement). Details about the sampling site, instrumentation and procedures for cloud sampling as well as methods for chemical analysis of cloud water samples are given in Deguillaume et al. (2014).

### 2.2 Isolation and identification of microorganisms from cloud waters

Triplicate volumes of 0.1 mL of cloud water are platted on R2A agar growth medium (Reasoner and Geldreich, 1985); DIFCOTM), and eventually also on R2A supplemented with NaCl 20 g L$^{-1}$, King's B (King et al., 1954), Sabouraud (DIFCOTM) and TSA (DIFCOTM) media. These are incubated at 17°C or 5°C under aerobic-dark conditions until appearance of colonies (typically 6 days at 17°C or 10 days at 5°C) (Vaïtilingom et al., 2012).

Representative colonies are selected on the basis of colony morphology and pigment production. Isolates obtained in pure cultures (R2A, 17°C) are stored in 10 % (v/v) glycerol at – 80°C. Strains are identified by ribosomal RNA gene sequencing (16S or 26S rRNA gene sequences for bacteria and yeasts, respectively). Full description of the methods of identification is available in Vaïtilingom et al. (2012).

### 2.3 Surface tension measurements

Strains from glycerol stock are used to inoculate R2A broth in 96 deep-well plates (500μL / well). Plates are incubated at 17 °C under agitation for 5 days, and then centrifuged (3000g / 20 min). Supernatants are





transferred into 1 mL microtubes and stored at -30 °C until surface tension measurements. Thawed samples are centrifuged (12,500 rpm / 3 min) just before surface tension measurements.

All surface tension measurements are performed using pendant drop method with an OCA 15 Pro tensiometer (Data Physics, Germany). The camera analyzes the pendant drop profile of crude extract. A dosing needle of 1.65 mm outside diameter is used, producing drops of 12 µL. The software fits this latter to the Young-Laplace equation and averages out surface tension from all measurements (Hansen and Rødsrud, 1991). Measurements are carried out at 295 K every second. The tensiometer is calibrated with Milli-Q water. The uncertainty of the instrument is ± 0.01 mN m$^{-1}$. Each dynamic surface tension curve is measured three times for the most efficient biosurfactant-producing microorganisms and the measurements display ± 10% of variation. These dynamic surface tension measurements last until the equilibrium region is reached (maximum 30 min, see below Section III.2). Along with the surface tension, each measurement also provides a real-time monitoring of the droplet volume, thus allowing checking for evaporation. No significant evaporation (< 5%) is observed during the experiments (Fig. 2).

### 2.4   Statistical analyses

Herein, we investigate the differences, in terms of impact on the non-normally distributed surface tension, due to the origin of air mass and the chemical composition of clouds using the PAST software version 3.09 (Hammer et al., 2001).

Using a non-parametric method, the Kruskal-Wallis one-way analysis of variance (Siegel, 1956), we compare the distributions of surface tensions between 4 air mass origin sectors: west (W), north-west/north (NW/N), north-east (NE) and south-west/south (SW/S) and between 4 chemical composition groups (Marine, Highly marine, Continental and Polluted). P-value < 0.05 is considered statistically significant.

Mann-Whitney test (Mann and Whitney, 1947), which is a measure of how different two populations are, allows specifying which group dominates, with two-by-two comparison.



## 3   Results

### 3.1   Identification of cloud microbial isolates

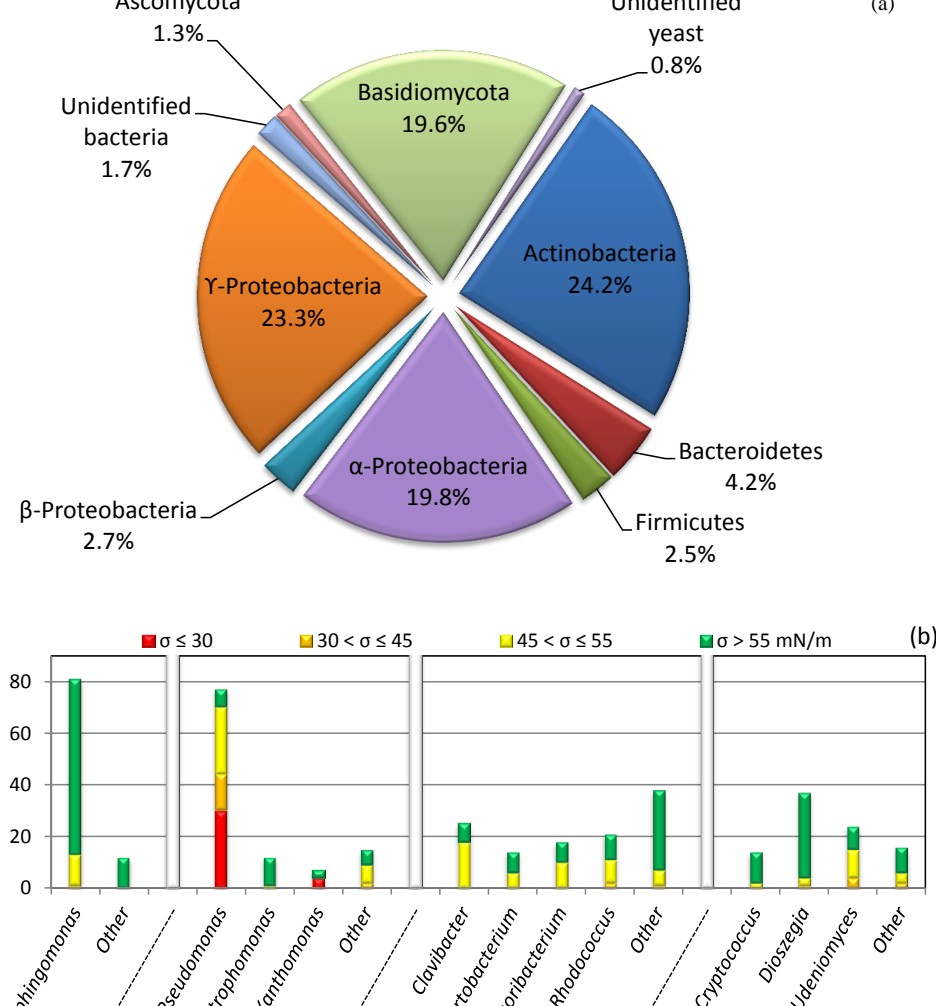

**Figure 1. (a)** Phyla distribution of 480 strains tested for biosurfactant production. **(b)** Genera distribution for most representative strains (85%: α- and ϒ-Proteobacteria, Actinobacteria and Basidiomycota). In red, orange, yellow and green, the 4 categories of surface tensions (σ ≤ 30, 30 < σ ≤ 45, 45 < σ ≤ 55 and σ > 55 mN m$^{-1}$, respectively).





Identifications of the 480 strains (bacteria and yeasts) collected during 39 cloud events at the puy de Dôme station, together with the values of surface tension obtained from their crude supernatants after 5 days of culture in R2A broth, are fully described in the Supplement (Table S2).

Figure 1(a) presents the distribution of the different phyla or classes of these microbial isolates. Three phyla of microorganisms dominate: Proteobacteria (α, β and ϒ- Proteobacteria), Actinobacteria and Basidiomycota, accounting for 89.6% of the collection, while 2.5% of the latter remain unidentified.

In detail (Fig. 1b), the phylum of Proteobacteria is predominant (220 isolates, 45.8%), notably α- and ϒ-Proteobacteria (95 and 112 isolates, 19.8% and 23.3% respectively). In these latter two classes, the most recurrent strains belong to the genera *Sphingomonas* (83 isolates) and *Pseudomonas* (78 isolates), respectively. β-Proteobacteria represent only 2.7% of the total. The phylum Actinobacteria (116 isolates, 24.2%) contains mainly strains from the genera *Clavibacter* (25 isolates), *Curtobacterium* (14 isolates), *Frigoribacterium* (18 isolates) and *Rhodococcus* (21 isolates). It is worth noting that the phylum Actinobacteria presents a much greater diversity of genera compared to the other phyla, centered on one dominant genus. Amongst bacterial strains, the phyla Bacteroidetes and Firmicutes are also represented but in a lesser extent (20 and 12 isolates respectively, 4.2% and 2.5%).

Concerning yeasts, the major group belongs to the phylum Basidiomycota (94 isolates, 19.6%) which contains mainly strains from the genera *Cryptococcus* (14 isolates), *Dioszegia* (39 isolates) and *Udeniomyces* (25 isolates). The phylum Ascomycota is also present but with only 6 isolates (1.2%).

Globally the phylogeny of the isolated strains used for this study is very consistent with that already published for strains isolated from the same sampling site, except for the genus *Bacillus* which is much less abundant in these selected clouds events (Vaïtilingom et al., 2012). The major point is that many strains originate from the phyllosphere as attested by the predominance of the Proteobacteria phylum. These 480 strains constitute therefore a unique collection of cloud microorganisms, representative of cloud community and that can be tested for their ability to produce biosurfactants.





### 3.2   Screening for biosurfactant-producing microorganisms

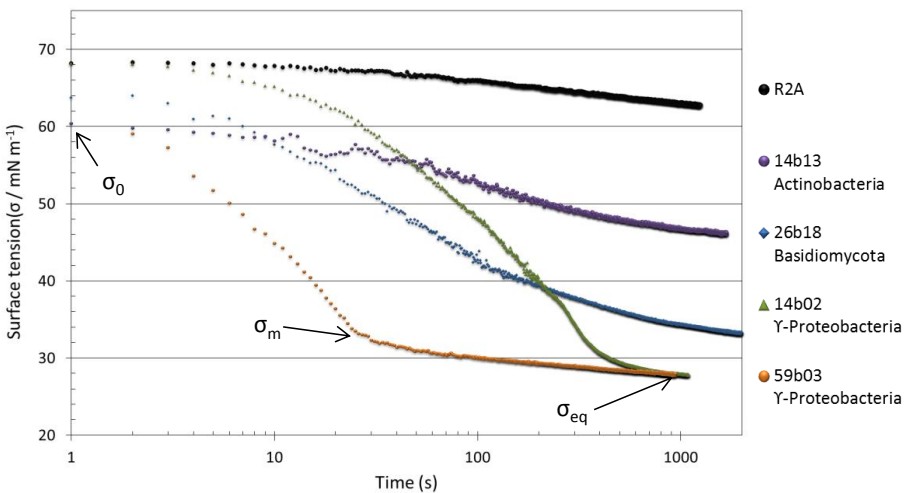

**Figure 2.** Time profile of surface tension measurements. In black, R2A broth medium. In purple, strain 14b13
(*Frigoribacterium* sp., Actinobacteria). In blue, strain 26b18 (*Cryptococcus* sp., Basidiomycota). In green, strain 14b02
(*Pseudomonas* sp., ϒ-Proteobacteria). In orange, strain 59b03 (*Pseudomonas syringae*, ϒ-Proteobacteria).

On fig. 2 are represented, as their time profile, surface tension measurements performed on the culture
supernatant of 4 selected microbial strains. As a reference, is also presented the surface tension obtained from
R2A medium, which served as culture medium. In this case, the observed surface tension (63 mN m$^{-1}$) remains
close to the value of pure water (72.8 mN m$^{-1}$). As expected, a more or less strong lessening of the surface
tension values is observed when microorganisms are able to produce biosurfactants in the culture medium. This
phenomenon is time-depending, time profiles depending themselves on the studied strain. These profiles are
consistent with those obtained from atmospheric aerosol by Nozière et al. (2014) and are typical of a surface
tension dynamic (Hua and Rosen, 1991). Indeed, three distinct kinetic regimes follow each other during the
equilibration process: first, a rapid fall from the R2A value to $\sigma_0$ occurs, too fast to be captured by our
instrument (< 0.1 s), then is observed a meso-equilibrium phase during which the surface tension decreases to $\sigma_m$
before the appearance of the equilibrium region where the minimum, $\sigma_{eq}$, is reached. This region corresponds to
the saturation of the surface ($\Gamma_\infty$) with surfactant molecules. Hereafter, surface tension measurements ($\sigma$) refer to
$\sigma_{eq}$.

Monitoring surface tension over time reveals that the equilibration time (apparent diffusion coefficient) in
pendant drops varies from few seconds to 30 min, probably depending on the concentration and the chemical
structure of the expressed biosurfactants that can affect molecular interactions and/or diffusion. For example,
from the supernatant of incubation media of the strains 59b03 and 14b02, two *Pseudomonas* strains (ϒ-
Proteobacteria), the measured equilibrium surface tensions ($\sigma_{eq}$) are close (below 28 mN m$^{-1}$), while time profiles
are much different and equilibration stage appears in around 2 and 10 minutes, respectively.





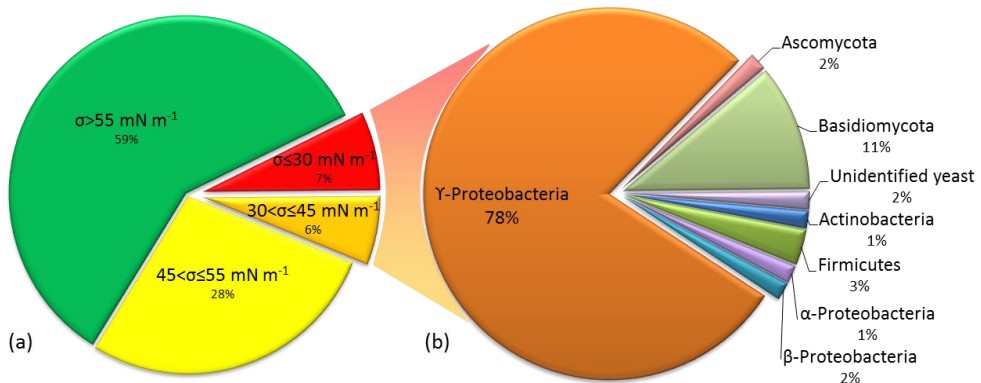

**Figure 3. (a)** Surface tension (σ) distribution of the 480 strains tested for biosurfactant production and **(b)** the phylum distribution for the most efficient biosurfactant-producing microorganisms ($\sigma \leq 45$ mN m$^{-1}$).

The 480 strains tested for biosurfactant production (See in the Supplement, Table S2), are differentiated into four main categories, according to the measured surface tension ($\sigma \leq 30$, $30 < \sigma \leq 45$, $45 < \sigma \leq 55$ and $\sigma > 55$ mN m$^{-1}$, Fig. 1b and Fig. 3). These 4 categories are chosen in a similar way to Baduel et al. (2012) and Ekström et al. (2010). The first category ($\sigma \leq 30$ mN m$^{-1}$) is rare among man-made surfactants and typical from surfactants of biological origin (Christofi and Ivshina, 2002). In our collection, we observe 34 strains (7%) able to reduce the surface tension of pure water below 30 mN m$^{-1}$. They exclusively belong to the genera *Pseudomonas* and *Xanthomonas* (ϒ-Proteobacteria, Fig. 1b). The second category corresponds to surface tension values up to 45 mN m$^{-1}$. This latter limit is often considered as the threshold in term of surface tension decrease originating from HULIS (humic like substances) (Kiss et al., 2005; Taraniuk et al., 2007). We observe only 30 strains (6%) in this second category. To sum up, from these two first categories ($\sigma \leq 45$ mN m$^{-1}$), and even though new phyla appears due to the second one, the phylum distribution of the most efficient biosurfactant-producing microorganisms remains largely dominated by ϒ-Proteobacteria (78% of all strains) and more moderately by Basidiomycota (11%) (Fig. 3). It must be emphasized here that the two others major taxa of all studied strains, Actinobacteria and α-Proteobacteria, almost completely disappear in these categories. The third and last categories ($45 < \sigma \leq 55$ and $\sigma > 55$ mN m$^{-1}$) represent 28 and 59 % of our collection, respectively. Outstandingly, it has to be noted that *Pseudomonas* (ϒ-Proteobacteria) and *Sphingomonas* (α-Proteobacteria) which are the most frequently found genera in clouds (Vaïtilingom et al., 2012), behave completely differently. Thus, *Pseudomonas* provide the most active biosurfactant-producing microorganisms while almost all *Sphingomonas* are not efficient for the production of biosurfactants.





### 3.3 Impact of the origin and chemical composition of clouds on biosurfactant production

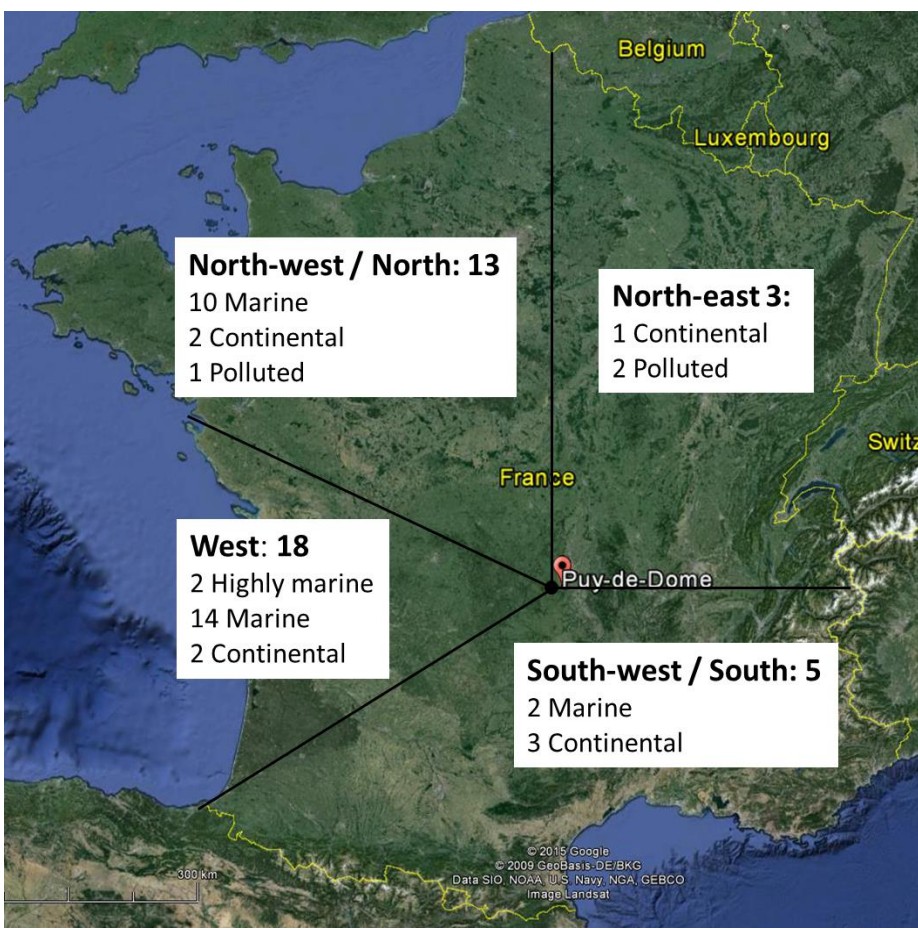

**Figure 4.** Air mass origin distribution of cloud events (W, NW/N, NE and SW/S) evaluated from their backward trajectories (See in the Supplement, Fig. S1) and physicochemical characteristics of cloud waters (Marine, Highly marine, Continental and Polluted) as described by Deguillaume et al. (2014) and summarized in the Supplement (Table S1).

In our study, the screened microbial strains are isolated from 39 cloud events presenting different profiles. The cloud chemical composition as well as the physicochemical parameters, measured at the puy de Dôme station and described in Deguillaume et al. (2014), can be found on the website of the Observatory of Earth Physics in Clermont-Ferrand (http://wwwobs.univ-bpclermont.fr/SO/beam/data.php). The main parameters are summarized in the Supplement (Table S1).

Cloud events are first classified according to the air mass origin - *i.e.*, west (W), north-west/north (NW/N), north-east (NE) and south-west/south (SW/S) - determined from their backward trajectories (see Fig. 4 and Fig. S1 in the supplement). Second, cloud events are classified according to the physicochemical characteristics of cloud waters (Marine, Highly marine, Continental and Polluted, see Fig S1 and Table S1 in the supplement) as described by Deguillaume et al. (2014).





Figure 4 shows that 18 events come from W consisting of 2 Highly marine, 14 Marine and 2 Continental cloud events, 13 events from NW/N with 10 marine, 2 continental and 1 polluted cloud events, 3 events from NE with 2 polluted cloud events and the other continental and, from the SW/S, 5 events of which 2 marine and 3 continental cloud events.

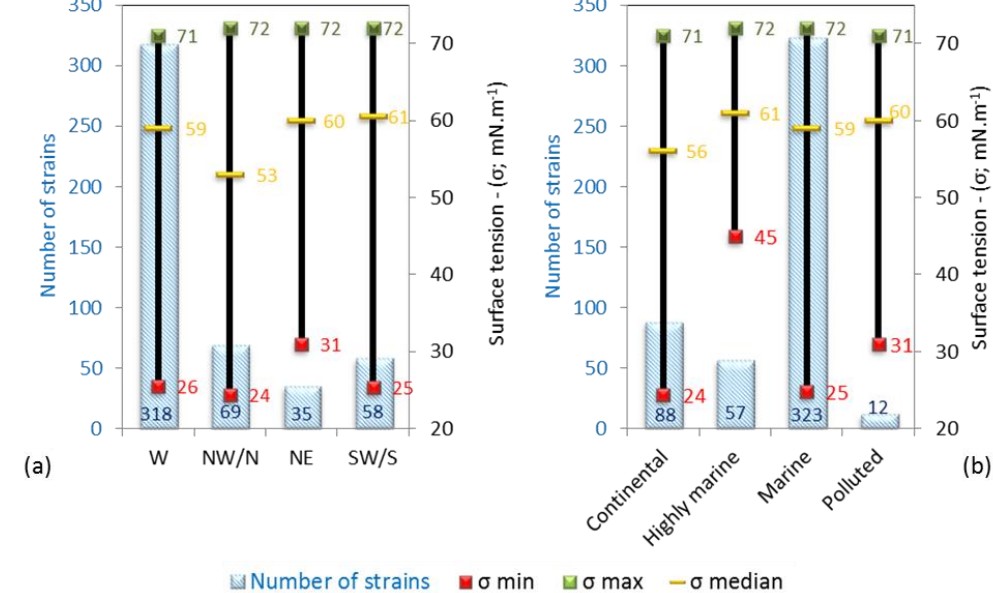

(a)                                                                                              (b)

**Figure 5.** Surface tension (σ) distribution of the 480 strains tested for biosurfactant production **(a)** according to air mass origin (4 sectors: W, NW/N, NE and SW/S), **(b)** according to physicochemical characteristics of cloud waters (Marine, Highly marine, Continental and Polluted). In blue, number of strains tested according to air mass origin. In red, orange and green, respectively, minimal, median and maximal surface tensions according to air mass origin.

Figure 5a shows the distribution of surface tensions values (σ) measured from the 480 strains tested for biosurfactant production, according to the air mass origins (4 sectors: W, NW/N, NE and SW/S). Samples from west sector constitute the great majority of our collection (318/480 strains). From statistical analysis, we observe a significant difference (Kruskal-Wallis p-value: 0.0049 << 0.05) in the distribution of biosurfactant-producing microorganisms between the four sectors (W, NW/N, NE and SW/S). The Mann-Whitney test (see details in the Supplement, Table S3) allows us to attribute this difference to the NW/N sector with a surface tension median (53 mN m$^{-1}$) significantly lower (Mann-Whitney p-values < 0.05) than the other three sectors (medians: 59, 60 and 61 mN m$^{-1}$, for W, NE and SW/S sectors, respectively). This difference cannot be completely attributed to the differences in the phyla distribution within the different air masses (Fig. 6). Indeed, as shown before, the most efficient biosurfactant-producing microorganisms belong to ϒ-Proteobacteria class, which represents 23% of all strains, but its distribution regarding the air mass origin sectors remains unselective (26, 28, 3 and 14% for W, NW/N, NE and SW/S sectors, respectively). The difference in the distribution of biosurfactant-producing microorganisms between the four sectors is rather due to the proportion of the most efficient biosurfactant-producing microorganisms (σ ≤ 45 mN m$^{-1}$) amongst the ϒ-Proteobacteria class (Fig. 6). In the NW/N sector, most efficient biosurfactant-producing microorganisms account for 68% of ϒ-Proteobacteria (13/19 isolates),



against 40% in the other three sectors (37/93 isolates). No such difference amongst the Υ-Proteobacteria class is observed in the chemical composition groups.

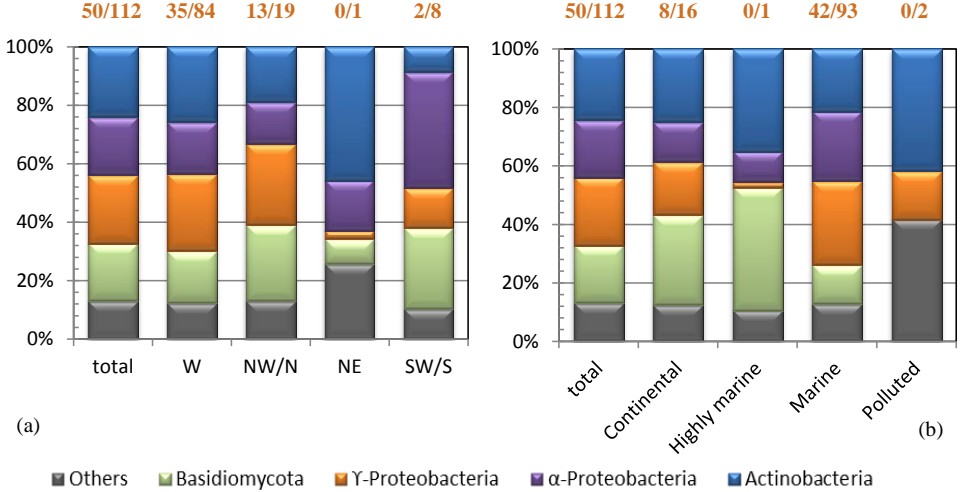

**Figure 6.** Phyla distribution according to **(a)** the air mass origin sectors: west (W), north-west/north (NW/N), north-east (NE) and south-west/south (SW/S); **(b)** the chemical composition groups (Marine, Highly marine, Continental and Polluted). P-value < 0.05 is considered statistically significant. In orange, highlighted the proportion of the most efficient biosurfactant-producing microorganisms (i.e., $\sigma \leq 45$ mN m$^{-1}$) amongst the Υ-Proteobacteria class (*e.g.*, In the NW/N sector, most efficient biosurfactant-producing microorganisms account for 68% of Υ-Proteobacteria, *i.e.*, 13/19 isolates).

Focusing on the impact of the cloud water chemical composition (Marine, Highly marine, Continental and Polluted, Fig. 5b) over surface tension values, a slight difference is also observed but the correlation is less significant (KW p-value: 0.025 < 0.05). According to the Mann-Whitney test (p-value: 0.01 < 0.05; see details in the Supplement, Table S3), this discrimination is mainly due to the difference between continental and highly marine strains (medians: 56 and 61 mN m$^{-1}$, respectively). Indeed, highly marine strains lead to a highest minimal surface tension (45 mN m$^{-1}$, Fig. 5b), minimal value in agreement with an almost total absence of Υ-Proteobacteria (1/57 isolates, see Fig. 6b).

## 4    Discussion and conclusion

The study of biosurfactant production in the environment has been mainly focused on microorganisms isolated from the soils, the rhizosphere and the phyllosphere, or the marine environment (Bodour et al., 2003; Jackson et al., 2015; Raaijmakers et al., 2010; Satpute et al., 2010). Concerning atmospheric waters, Ahern et al. (2007) were the only ones to report the presence of biosurfactant-producing bacteria in the air, studying 100 strains isolated from 4 rain and cloud samples. Here, we investigate 480 strains isolated from 39 different cloud events. If we consider that strains are clearly producing biosurfactants when the surface tension measured ($\sigma_{eq}$) is under 55 mN m$^{-1}$, 41% of the tested strains are active, 7% being extremely active ($\sigma < 30$ mN m$^{-1}$). Although the methods used to evaluate biosurfactant production are different, this result is in agreement with that of Ahern et



al. (2007) who found 55% of active strains in their rain and cloud samples. In our study, we have shown that from our microorganism collection, the most efficient biosurfactant-producing microorganisms ($\sigma < 45$ mN m$^{-1}$) belong to a limited number of bacterial genera (*Pseudomonas* and *Xanthomonas*) from the ϒ-Proteobacteria class (78%), and yeast genus (*Udeniomyces*) from the Basidiomycota phylum (11%). Some *Bacillus* strains from the Firmicutes phylum are also active but represent a very small fraction (3%) of the total population of our cloud collection. Strains from the Actinobacteria phylum are mainly present in the group with a moderate biosurfactant production ($45 < \sigma < 55$ mN m$^{-1}$). In the literature major bacterial genera reported as high biosurfactant-producing microorganisms concern *Pseudomonas* (ϒ-Proteobacteria) and *Bacillus* (Firmicutes*)*; *Acinetobacter* ($\gamma$-Proteobacteria) is also frequent (Desai and Banat, 1997; Rosenberg and Ron, 1999). For the yeasts the major genera cited are *Candida* and *Torulopsis* from Ascomycota phylum (Desai and Banat, 1997; Karanth et al., 1999; Rosenberg and Ron, 1999). In our study *Pseudomonas* strains are clearly the dominant group and the most active biosurfactant-producing microorganisms, but *Acinetobacter genus* is absent. This result is highly consistent with studies performed on environmental samples such as soils (Bodour et al., 2003), plants (D'aes et al., 2010; Raaijmakers et al., 2010), seawaters (Cai et al., 2015) and in atmospheric waters (Ahern et al., 2007).

It is important to note that *Pseudomonas* genus is commonly found in the phyllosphere which is accepted as the main source of primary bioaerosols (Amato et al., 2007b; Morris et al., 2014; Vaïtilingom et al., 2012). Interestingly it has been shown that biosurfactants play a specific role in the interaction between plants and *Pseudomonas* (D'aes et al., 2010; Raaijmakers et al., 2010). Biosurfactants present versatile functions including interactions with other organisms (such as antibiotic activity) and modification of the leaf-surface properties. This surface modification allows cell mobility, biofilm formation and thus the colonization of the leaves by these bacteria. In particular rhamnolipids are involved in the different processes of biofilm formation; the final step consists in releasing the planktonic daughter population (Mann and Wozniak, 2012). This production of biosurfactants could be therefore of major importance for biofilms present on leaf surfaces, allowing the aerosolization and dispersion of *Pseudomonas* strains in the air.

This aerosolization of *Pseudomonas* strains could explain the correlation observed between the origin and composition of clouds with the distribution of biosurfactant-producing microorganisms observed in our study. It is clear that microbial isolates from highly marine clouds which are greatly impacted by the ocean source (an almost total absence of ϒ-Proteobacteria, see Fig. 6) are lower biosurfactant producers than microorganisms isolated from continental clouds. These latter, travelling over vegetated zones, contain thus more *Pseudomonas* strains (see Fig. 4). More generally, the correlation between the different origins of the air masses and the production of biosurfactants by cloud microorganisms could be explained by the great differences of vegetation of France. For instance the predominance of most efficient biosurfactant-producing microorganisms in the clouds originating from the Northwest-North sector could be justified by agricultural practices. These French regions are characterized by uniform monocultures (Ministère de l'Agriculture, de l'Agroalimentaire et de la Forêt, 2014) extremely favorable to plant pathogens, such as *Pseudomonas syringae* (McDonald and Linde, 2002).



The occurrence of biosurfactants has been recently shown on atmospheric aerosols (Baduel et al., 2012; Ekström et al., 2010; Nozière et al., 2014). Actually the presence of biosurfactants in the atmosphere could result from different mechanisms. First they could be directly produced by microorganisms in clouds. This is relevant considering that microorganisms have been shown to be metabolically active in clouds and can survive under atmospheric conditions. Indeed, microbial metabolic activity has been demonstrated by measurements of the ATP (adenosine 5'triphosphate) content (Amato et al., 2005, 2007a; Vaïtilingom et al., 2012, 2013) and by the uptake of the dye CTC (5-cyano-2,3-ditolyl tetrazolium chloride) directly *in situ* in cloud water samples (Hill et al., 2007). This activity have been proved even at low temperatures (Sattler et al., 2001). In addition these microorganisms are able to survive and resist to atmospheric stresses including for instance evaporation-condensation cycles, freeze-thaw cycles, exposure to oxidants and solar light and cold temperature (Delort et al., 2010; Joly et al., 2015). Biosurfactants can be also produced in extreme environment such as deep sea or Arctic soil (Jackson et al., 2015; Janek et al., 2013). In our laboratory conditions, microorganisms were able to produce biosurfactants within the first 24 hours, even in poor nutritive medium (R2A). As the residence time of microorganisms in the atmosphere was modeled to be between 2 to 10 days (Burrows et al., 2009), these metabolically active microorganisms could thus synthesize biosurfactants in clouds. A second and obvious pathway for the incorporation of biosurfactants in atmosphere is related to their presence at the surface of the microorganisms. They can therefore be carried together with the microorganism when aerosolized or in a biofilm existing on dust or leave particles. Finally biosurfactants could be directly emitted in the atmosphere as biogenic aerosols, this could be particular true for biosurfactants of marine origin that can be emitted in sea sprays during bubbling and wave braking processes (Blanchard, 1989; Elliott et al., 2014).

The presence of biosurfactants might have implications for atmospheric processes: First it could impact atmospheric chemistry. For instance *in situ* biosynthesis of biosurfactants by microorganisms in clouds could be considered as the production of secondary organic aerosols, modifying thus the global carbon balance in the water phase. The presence of biosurfactants (aerosolized from the earth surface or produced *in situ)* at the surface of cloud droplets could also change the actual picture of mass transfer between the gas and water phases of clouds. Organic surface films can provide barriers to transport across the air−particle interface, inhibiting uptake of water and gas phase species. These organic films can be an auspicious medium for solubilizing gas phase organic species that may perhaps explain the observed non-Henry's law concentration of organics in field samples (Barnes, 1986; Davies et al., 2013; Lo and Lee, 1996; Park et al., 2009; Renard et al., 2014). The chemical characterization and the study of the reactivity of these organic layers will be of special interest to go further in understanding cloud chemistry.

Second, biosurfactants could affect atmospheric microphysics by modifying CCN activation. Owing to their exceptional scope in reducing surface tension, biosurfactants *per se*, present on aerosols or associated at the microorganisms surface, are thus likely to enhance the propensity of these aerosols to form clouds as activation of particles into cloud droplets depends on surface tension according to the Köhler's theory (Köhler, 1936). This topic has been controversial but recently Nozière et al. (2014) showed that the total surfactant fraction of atmospheric particles is much more surface-active ($\sigma \leq 30$ mN m$^{-1}$) than HULIS for example. They demonstrated that the equilibration time of biosurfactants might have hindered the measurement of such an effect when using





classical instrument such as hygroscopic tandem differential mobility analyzer and CCN counter. Looking in more details to our results one could argue that biosurfactant partitioning in macroscopic pendant droplets may decrease our surface tension values relative to atmospheric conditions. Indeed, in microscopic ($D_{wet} \approx 1\ \mu m$) droplet, the partitioning impact could be insignificant owing to a surface area to volume ratio several orders of magnitude greater (Prisle et al., 2012; Sorjamaa et al., 2004). Nevertheless, lower is the radius of the droplet or higher is the surface to volume ratio, higher is the concentration of bacteria (Aller et al., 2005; Hejkal et al., 1980) and higher is the concentration of WSOC (Ervens and Volkamer, 2010). By dividing the surfactant concentration in the atmosphere ($\sim 10^{-12} - 10^{-9}$ mol m$^{-3}$ in (Olkowska et al., 2014) by the liquid water content of wet aerosol ($\sim 10^{-6} - 10^{-5}$ g m$^{-3}$ in Ervens and Volkamer, 2010), we would obtain significant concentration of $\sim$mM. Recently, Gérard et al. (2016) observed such concentrations, that would be above typical CMC. Thus, Ruehl et al. (2012) presented strong evidence that surface tension reduction can occur in microscopic droplets, and even more in wet aerosol, provided that particles are predominantly (*i.e.*, $\gtrsim 80\%$) composed of surfactant.

The influence of biological surfactants on the prediction of particle cloud activation and aerosol indirect climate effects need to be implemented in models. Indeed, if the effects of organic surfactants (mainly carboxylic acids) on the surface tension of activating droplets have been considered in parameterizations such as the one from Abdul-Razzak and Ghan (2004), recent studies have however shown that the surface partitioning of organic molecules to a microscopic aqueous droplet interface have to be also considered in models (Noziere, 2016; Prisle et al., 2012; Ruehl et al., 2016).

Finally because *Pseudomonas* strains are the most efficient biosurfactant-producing bacteria and are dominant in clouds and rain (this work and Ahern et al., 2007), the question rises about the potential role of biosurfactants in the cycle of *Pseudomonas* in the atmosphere. Indeed, biosurfactants help in the aerosolization of theses strains from leaf surfaces then favor the formation of cloud droplets by modifying the surface properties of the cells. Moreover, most of these strains belong to the *Pseudomonas syringae* species that are known to be ice nuclei active and can induce precipitations back to the earth. Biosurfactants should thus be integrated in the life history of *Pseudomonas syringae* and its linkage to the water cycle (Morris et al., 2008, 2014).

To our knowledge, research on the potential impact of biosurfactants on human health due to their presence in atmospheric waters remains marginal, compared to terrestrial or aquatic ecosystem studies (Olkowska et al., 2014). Aerosols are now well-known to represent a major concern for the populations as shown by epidemiology studies (Bernstein et al., 2004; Dominici et al., 2014; Pope, 2000). The toxicological impact is inversely proportional to the particle size (Nel, 2005; Novák et al., 2014). For example, fine particles ($PM_{1-2.5}$) reach lung alveoli and ultrafine particles ($PM_{0.1}$), once in the lung, would readily pass into the bloodstream and cause a direct insult to the cardiovascular system and other organs (Moshammer and Neuberger, 2003; Polichetti et al., 2009). The composition of the aerosols is also of major importance and should not be underestimated (Brimblecombe and Latif, 2004; Škarek et al., 2007). Regarding more precisely surface-active organic aerosols, few reports are devoted to synthetic molecules. Thus, Poulsen et al. (2000) suggested that molecules with surfactant properties could interfere in the immunological pathways, which could explain the increase of allergic diseases in industrialized societies. At high concentration in the atmosphere, surfactants can lead to asthma, can disrupt the stability of human respiratory systems, as well as can cause dry eyes allergies (Ahmad et al., 2009;



Xinxin et al., 2016). Rhamnolipids inhibit ciliary function and produce damage to the human bronchial epithelium (Abdel-Mawgoud et al., 2011). Surfactants, by lowering the surface tension of tear films of the eyes could also be at the origin of dry eyes sensation (Vejrup and Wolkoff, 2002). Hence, biosurfactants present on aerosols could have a double impact on human cells, first because they could destroy cell membranes of the host, second because they could concentrate and dissolve toxic pollutants and help their penetration within the host cells. Further studies are needed to better evaluate the impact of surfactant on human health.

In conclusion, our work shows that microbial strains isolated from cloud waters are able to produce strong biosurfactants. The major and most active producers belong to the *Pseudomonas* genus which is prevalent in cloud waters and originates commonly from the phyllosphere. Although the presence of surfactants has been shown on aerosols, their structure has yet to be established. The biosurfactants overproduced by our best producers will be isolated in order to analyze their chemical structure. In parallel biosurfactants from aerosols samples will be also extracted and structural fingerprints will be analyzed and compared with the signature of microbial surfactants isolated from clouds. These comparisons should allow the evidence of the microbial origin of the surfactants present on aerosols. Studying such biosurfactants in the atmosphere is of special interest for the chemical characterization and reactivity of organic layers, their impact on mass transfer and water uptake. Activation of aerosols containing organic matter is a major topic directly related to aerosols-cloud-interactions climatic effects. A small change in droplet population could affect cloud albedo as well as the formation of precipitation (Li et al., 2011; Rodhe, 1999). Hence comes the need to enhance our knowledge about biosurfactants, whether they could impact human health and global climate, and in what proportion (Brimblecombe and Latif, 2004).

**Acknowledgements**: This work was funded by the French-USA program ANR-NSF SONATA and the French-Slovak Program Stefanik. Cloud sampling was possible thanks to the staff of the Observatoire de Physique du Globe de Clermont-Ferrand. Magali Abrantes is greatly acknowledged for her technical help.





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
