# Peer review of "Screening of cloud microorganisms isolated at the puy de Dôme (France) station for the production of biosurfactants"

_Atmospheric Chemistry and Physics, 2016_

## Referee Comment (RC1) · Anonymous Referee #1 · 24 Jun 2016

Renard et al. investigate the production of biosurfactants by microorganisms isolated from cloud water. This is accomplished by measuring the surface tension of the crude extracts (supernatants from pure cultures) using a pendant drop tensiometer. The authors find that several strains produce highly surface-active compounds. Most notably, Pseudomonas is one of the most abundant and efficient biosurfactant-producing microorganisms in their study. This work makes a meaningful contribution to the field; many groups have investigated both the effect of biosurfactants from terrestrial samples and the effect of surfactants on atmospheric waters, but only one other group has investigated the effect of biosurfactants on atmospheric waters. This may be publishable after the following issues are addressed.

[Figure]

The authors report equilibrium surface tension values for every microorganism in their study. However, it is unclear how they determine when equilibrium has been reached. All of the surface tension time profiles in Figure 2 appear to be decreasing when the measurements were stopped. That is, the reported equilibrium surface tension values are the minimum values for the time profiles given, but may not be if the time profile was extended. In section 2.3, the authors state a 30-minute maximum for surface tension measurements but give no justification for this time frame.

In Figure 2, the authors show the surface tension time profile for the R2A broth, which is the medium used for all cultures of their isolated microorganisms. However, this may not be a good baseline because incubation period lasts between six to ten days. We accept that the microorganisms are altering the composition of the broth by producing biosurfactants, but they are consuming nutrients in the broth as well. It remains to address how the removal of nutrients would impact the surface tension of the crude extracts.

Furthermore, the authors present a large amount of data regarding the surface tension of crude extracts but do not make a connection to the surface tension of cloud water, which is arguably the basis for this work. Since the authors have already collected the cloud water in order to isolate the microorganisms, it would be useful to also report surface tension values for the cloud water samples as well extend the crude extract results to cloud water.

Finally, the statistical analysis section did not seem to add much to the paper. The main takeaway was that $\gamma$-Proteobacteria are efficient biosurfactant producers, which reinforces conclusions from section 3.2. However, the entire analysis seems unsubstantiated. The distinction between air mass origins seems arbitrary. The distinction between chemical compositions is more logical, but the conclusions for that analysis are weaker.

The grammatical errors are too numerous to list individually. This paper would greatly

benefit from editing by a native English speaker.

Page 5, line 2. Might be helpful to keep units consistent with Page 4, line 33. Either g (preferred) or rpm.

Page 5, line 11. Change section number from Roman to Arabic numerals.

Page 9, lines 9-10. This is a misrepresentation because the biosurfactants are reducing the surface tension of the R2A broth, not pure water.

Page 9, line 11. I think you mean surface tension values between 30 and 45 mN m-1 not up to 45 mN m-1.

Page 9, line 18. Third and fourth is clearer than third and last.

Page 11, lines 12-14. There is not a significant difference between all four sectors, just between NW/N and the others, according to your supplementary information.

Page 14, lines 3-5. Citation for this sentence?
* * *

---

## Referee Comment (RC2) · Anonymous Referee #2 · 24 Jun 2016

This manuscript by Renard et al. isolated bacteria from cloud droplets and analyzed the bacteria's ability to produce biosurfactants. The authors identified surfactants by culturing the bacteria, centrifuging the samples, and using a pendant drop tensiometer on the supernatant to determine surface tension. The authors demonstrate that a number of biosurfactant-producing bacteria exist in their samples. In particular, the a- and ÏŠ-Proteobacteria phyla are shown to be widespread and effective biosurfactant producers. This work is novel and well within both the scope of ACP and the interests of its readers. However, the manuscript as it stands has some major outstanding issues that must be addressed before publication can be recommended.

Major Comments:

- English grammar: This paper must be completely proofread by a native English speaker before publication should be considered.

-Clouds: the largest hole in this manuscript is the lack of cloud-water analysis. The authors acceptably demonstrate that biosurfactant producing bacteria exist in their cloud water samples but fail to demonstrate if the bacteria actually have a measurable effect on their collected cloud droplets. Even if the authors are unable to measure surface tension depression in their cloud samples, this should still be noted and contextualized in the manuscript. The paragraph in the discussion section, starting P14-L33, would greatly benefit from this analysis.

-Arbitrary choices: While not debilitating to the paper itself, two arbitrary divisions are made in this manuscript: 1) the surface tension division of Fig.3 and 2) the source region division of Fig. 4

For the surface tension divisions, the authors cite Baudel et al. (2012) and Ekstrom et al. (2010) saying that their divisions are chosen in a similar way. However, neither Baudel nor Ekstrom divide their data in the same way that the authors here are trying to do. It would be more correct to say that the authors are choosing bins for their samples that match 1-2 bins of previously published works. The other bins are completely arbitrary and the authors do not provide any reasons for why they chose >55, 45-55, 30-45, and <30 mN/m. Some reasoning behind these bins needs to be present in the manuscript.

For the source region divisions, the authors present no reasoning for dividing the air masses into marine/highly marine/etc. From Table S1, it is impossible to tell why marine and highly marine are split. Fig. S1 suggests the split is because of time over open ocean but the authors need to be explicit here. I would also argue that Fig. 4 adds nothing to the manuscript and should be replaced with the HYSPLIT trajectories and some additional meteorological statistics (e.g. wind direction histogram for the sampling site for both cloud-sampling days and non-sampling days). There is also no

reason to not show the air mass height results from HYSPLIT.

Furthermore, the analysis starting on page 11, which attributes statistical difference between the air mass divisions (one of which has 3 samples!), seems weak. I would suggest that there are stronger ways to segregate the air masses given the Table S1. I believe the whole analysis should be rerun using the chemical speciation data.

Shorter Comments:

-Mann-Whitney and KW ANOVA – can the authors comment on how appropriate it is to assume statistical independence for air masses that, though they are measured at the sampling site as from 2 different directions, shared the same path as few as 5 days prior to sampling?

-Figure 5 needs to be replaced with a proper box and whiskers plot or some other way to judge what the real spread in the data looks like (possibly note the std. dev.?)

-Paragraph starting P15, L26 – I am generally a proponent of contextualizing your work but I am not convinced this paragraph adds anything to the manuscript. It should either be removed or condensed and moved to the introduction.

Minor comments:

- S2.2, P4, L23-25 : Given that most of the people reading ACP are not biologists, a one sentence explanation on why those three specific agars were chosen should be added to the text. Also, TSA should be defined.

-P5, L11: Should read "3.2" not "III.2"

---

## Author Comment (AC1) · 24 Aug 2016

Dear reviewer 1,

Please find below our answer and enclosed our revised manuscript entitled" Screening of cloud microorganisms isolated at the puy de Dôme (France) station for the production of biosurfactants", by P. Renard, I. Canet, M. Sancelme, N. Wirgot, L. Deguillaume, and A.-M. Delort that we would like to publish in Atmospheric Chemistry and Physics; as well as, the supplementary material, the new figures 2 and 4ab, and the answers to the reviewers.

Kind regards

[Figure]

Manuscript acp-2016-447 Screening of cloud microorganisms isolated at the puy de Dôme (France) station for the production of biosurfactants P. Renard, I. Canet, M. Sancelme, N. Wirgot, L. Deguillaume, and A.-M. Delort

Answer to Referee #1 First we would like to thank to the reviewers for their work and interest in our work. We have taken into account their comments to improve the manuscript and answered point by point to their questions. Changes in the manuscript are underlined in yellow.

Anonymous Referee #1 The authors report equilibrium surface tension values for every microorganism in their study. However, it is unclear how they determine when equilibrium has been reached. All of the surface tension time profiles in Figure 2 appear to be decreasing when the measurements were stopped. That is, the reported equilibrium surface tension values are the minimum values for the time profiles given, but may not be if the time profile was extended. In section 2.3, the authors state a 30-minute maximum for surface tension measurements but give no justification for this time frame.

Author's response You are absolutely right, the minimum of the equilibrium region ($\sigma$eq), is difficult to determine experimentally (small variations of surface tension over long timescales) (Nozière et al., 2014). The surface tension decreases asymptotically and the logarithmic scale of the figure 2 is probably misleading, it looks clearer when presented in a non-logarithmic form (see enclosed Figure 2. Time profile of surface tension measurements in a non-logarithmic form.). The overall equilibration time, teq, is of the order of 2 × tm (time of meso-equilibrium) (Nozière et al., 2014). After this period, surface tension decreases marginally. According to the measurements we did on longer period (few hours), our overestimation is comprised between 0.1 and 0.2 mN m-1. That is why, above 30 mN m-1, we only give nominal surface tension (Table S2 in the supplementary). Below 30 mN m-1 and above the CMC, surface tension decreases quickly and it is easier to be more accurate. Finally, we decided to keep our original Figure 2 presented in the logarithmic mode as it is the usual way in publications and allows to show the initial surface tension ($\sigma$0), the surface tensions in the

meso-equilibrium ($\sigma$m) and in the equilibrium ($\sigma$eq) phase (See Noziere et al 2014 for instance).

Anonymous Referee #1 In Figure 2, the authors show the surface tension time profile for the R2A broth, which is the medium used for all cultures of their isolated microorganisms. However, this may not be a good baseline because incubation period lasts between six to ten days. We accept that the microorganisms are altering the composition of the broth by producing biosurfactants, but they are consuming nutrients in the broth as well. It remains to address how the removal of nutrients would impact the surface tension of the crude extracts.

Author's response We agree that broth, i.e. carbon sources, influences biosurfactant production. This is true for industrial production of biosurfactants performed in aqueous media with a rich carbon source feedstock, such as carbohydrates, hydrocarbons, fats, and oils. Such enriched broths increase the production. However in our case R2A broth is very poor so when the microorganisms consume the carbon sources, it does not make a great change. We confirmed this point thanks to the following experiments: We did purification of few biosurfactants from microbial R2A cultures, and then we measured the surface tension of these pure compounds both in water and in R2A medium, the differences were marginal (confidential, to be published).

Author's changes We modified the text as follows: P 9 line 9: In this collection, we observed 34 strains (7%) that reduce the surface tension of the R2A broth below 30 mN m-1.

Anonymous Referee #1 Furthermore, the authors present a large amount of data regarding the surface tension of crude extracts but do not make a connection to the surface tension of cloud water, which is arguably the basis for this work. Since the authors have already collected the cloud water in order to isolate the microorganisms, it would be useful to also report surface tension values for the cloud water samples as well extend the crude extract results to cloud water.

Author's response We share your viewpoint; the surface tension of cloud water could have been relevant. Unfortunately, the cloud samplings have been performed before the acquisition of the tensiometer. However, according to the Köhler theory, the surface tension, as well as, the saturation vapor pressure and the CCN diameter, drive the activation of particles into cloud droplets. The activation occurs when the radius of the cloud droplet is minimal (few nm, i.e., wet aerosol) and the concentration of organic compounds, such as biosurfactant, is maximal (Nozière et al., 2014). The effect of surface tension is maximal during the activation. In cloud droplet (few $\mu$m), organic compounds are diluted, and biosurfactants are likely under the CMC. Nevertheless, measuring surface tension in concentrated cloud water could be a complementary work, especially since we observed in 300 fold-concentrated rain, a strong decrease of the surface tension (30 mN m-1) (unpublished data). Here, we demonstrate that bacteria sampled in clouds are able to produce biosurfactants under lab conditions. We are currently isolating and characterizing these biosurfactants. We have identified 11 different structures by mass spectrometry. In the future we want to collect cloud and rain samples and also aerosols and look for these structures in these atmospheric samples (this is what is proposed in the conclusion). This is a long term research plan.

Author's changes In order to emphasize this point, we have modified the following sentences: P 13 lines 4: "In conclusion, the results of the present study showed that the microbial strains isolated from cloud waters produce strong biosurfactants under laboratory conditions. The major and most active producers belong to the Pseudomonas genus, which is prevalent in cloud water and typically originates from the phyllosphere. Although the presence of surfactants has been shown on aerosols (Nozière et al., 2014), it has not yet been demonstrated in clouds, and the structure of these compounds has not been established. The biosurfactants overproduced by the best producers in the present study will be isolated to analyze their chemical structure. In parallel, the biosurfactants from cloud aerosols and rain samples will also be extracted, and their structural fingerprints will be analyzed and compared with the signatures of microbial surfactants isolated from clouds."

Anonymous Referee #1 Finally, the statistical analysis section did not seem to add much to the paper. The main takeaway was that $\alpha$-Proteobacteria are efficient biosurfactant producers, which reinforces conclusions from section 3.2. However, the entire analysis seems unsubstantiated. The distinction between air mass origins seems arbitrary. The distinction between chemical compositions is more logical, but the conclusions for that analysis are weaker.

Author's response Our statistics are based on 480 strains but these strains are grouped into 39 cloud events, thus partially dependent. This sampling was spread over 10 years, and represented with related analyzes, a considerable work and a more than correct observation of Puy de Dôme clouds. However it is still difficult to make statistics on samples with such intra- and inter-sample variations. For example, in marine clouds, we identified only one strain in few events (e.g., event 29) compared to the 62 strains in the event 54 (see Table S1 in supplementary). This makes our Mann-Whitney and Kruskal-Wallis tests a bit weak. We could use mixed model. Nevertheless, you are right, these statistics would not add much new, i.e., the correlation of Pseudomonas / surface tension. We therefore concluded it would be better to be limited to a high-quality observation.

In conclusion we decided to keep the paragraph "Impact of the origin and chemical composition of clouds on biosurfactant production" to give some general tendency. The obtained results are interesting as they suggest a link between the vegetation origin and the biosurfactant production. This should be studied in more details in the future.

Author's changes In the abstract, we replaced: Statistical analyses showed some positive correlations between the origin of air masses and chemical composition of cloud waters with the presence of biosurfactant-producing microorganisms, suggesting a "biogeography" of this production. by: We observed some correlations between the chemical composition of cloud water and the presence of biosurfactant-producing microorganisms, suggesting the "biogeography" of this production. Page 4 line 5: we replaced: "In order to evaluate the potential correlation between the origin of air masses and composition of cloud waters and the presence of biosurfactant-producing microorganisms, statistical analyses are performed." by: "We observed a potential correlation between the composition of cloud waters and the presence of biosurfactant-producing microorganisms." P5 line 14: This text has been deleted: 2.4 Statistical analyses Herein, we investigate the differences, in terms of impact on the non-normally distributed surface tension, due to the origin of air mass and the chemical composition of clouds using the PAST software version 3.09 (Hammer et al., 2001). Using a non-parametric method, the Kruskal-Wallis one-way analysis of variance (Siegel, 1956), we compare the distributions of surface tensions between 4 air mass origin sectors: west (W), north-west/north (NW/N), north-east (NE) and south-west/south (SW/S) and between 4 chemical composition groups (Marine, Highly marine, Continental and Polluted). P-value < 0.05 is considered statistically significant. Mann-Whitney test (Mann and Whitney, 1947), which is a measure of how different two populations are, allows specifying which group dominates, with two-by-two comparison. Page 10 line 1: we totally rewrote the section 3.3 and replaced by: 3.3 Potential impact of the chemical composition of the clouds on biosurfactant production In the present study, the screened microbial strains were isolated from 39 cloud events presenting different profiles. Information on the cloud chemical composition and the physicochemical parameters measured at the puy de Dôme station and described in (Deguillaume et al., 2014) is provided on the website of the Observatory of Earth Physics in Clermont-Ferrand (http://wwwobs.univ-bpclermont.fr/SO/beam/data.php). The main parameters, including pH, $SO_4^{2-}$, $NO_3^-$, $Cl^-$, acetate, formate, oxalate, succinate, malonate, $Na^+$, $NH_4^+$, $Mg^{2+}$, $K^+$, and $Ca^{2+}$, are summarized in the Supplemental materials (Table S1). These physico-chemical parameters were used for the ACP analysis as described in Deguillaume et al. (2014). The ACP generated 4 different types of clouds, classified as "highly marine", "marine", "continental" and "polluted". Typically, the more "polluted" clouds have a lower pH and higher concentrations of $NH_4^+$, $NO_3^-$, and $SO_4^{2-}$. The more "marine" clouds have a higher concentration of NaCl. The 39 cloud events were

divided into 2 "highly marine", 26 "marine", 8 "continental" and 3 "polluted" clouds (Table S1).

Figure 4 (a). Surface tension ($\sigma$) distribution of the 480 strains examined for biosurfactant production according to the physicochemical characteristics of cloud waters (marine, highly marine, continental and polluted). Highlighted in blue, the number of tested strains. Box and whisker plots are shown with the minimal (red) and maximal (green) surface tensions. The orange boxes represent the 25th and 75th percentiles of the measurements (b). Phyla distribution according to the physicochemical characteristics of the cloud waters (marine, highly marine, continental and polluted). Figure 4a shows the distribution of the surface tensions values ($\sigma$) measured from the 480 strains examined for biosurfactant production according to the cloud water chemical composition (marine, highly marine, continental or polluted). A comparison of the distribution of the phyla of the strains in the same cloud events is presented in Figure 4b. The samples from marine clouds constitute the majority of this collection (323/480 strains). We observed a difference between the surface tension values from continental and highly marine strains (medians: 56 and 61 mN m-1, respectively). Highly marine clouds are characterized by the highest minimal surface tension (45 mN m-1, Figure 4a), consistent with the almost complete absence of ÏŠ-Proteobacteria, which are the most efficient biosurfactant-producing microorganisms ($\sigma \leq$ 45 mN m-1) (1/57 isolates, see Figure 4b). These observations were based on 39 cloud events with 480 different strains, representing, to our knowledge, the largest cloud sample data set studied; this data set is representative of cloud sampling over more than 10 years at the puy de Dôme station. Although it remains difficult to generate statistics on samples with such intra- and inter-sample variations, these results provide a general tendency that could be reinforced and confirmed with more data in the future. Figure 4, Figure 5a, Figure 6a, Table S3 and Figure S1 have been deleted. We have kept only Figure 5b and Figure 6b which are now Figure 4(a) and (b) in the revised manuscript, note that the presentation of the data has been modified as suggested by referee 2.

Anonymous Referee #1 The grammatical errors are too numerous to list individually. This paper would greatly benefit from editing by a native English speaker. Authors: The manuscript has been proof read by ACS services Page 5, line 2. Might be helpful to keep units consistent with Page 4, line 33. Either g (preferred) or rpm. Authors: We put these values: 10,480 g / 3 min Page 5, line 11. Change section number from Roman to Arabic numerals. Authors: done Page 9, lines 9-10. This is a misrepresentation because the biosurfactants are reducing the surface tension of the R2A broth, not pure water. Authors: Changed Page 9, line 11. I think you mean surface tension values between 30 and 45 mN m-1 not up to 45 mN m-1. Authors: Yes, we agree, we have changed it to "between 30 and 45 mN m-1" Page 9, line 18. Third and fourth is clearer than third and last. Authors: Changed Page 11, lines 12-14. There is not a significant difference between all four sectors, just between NW/N and the others, according to your supplementary information. Authors: Actually we have deleted all the data linked to the back-trajectories of the air masses (see answer to referee 2) Page 14, lines 3-5. Citation for this sentence? Authors: we have added "(Joly et al., 2015 and references hereinafter)".

Please also note the supplement to this comment:
http://www.atmos-chem-phys-discuss.net/acp-2016-447/acp-2016-447-AC1-supplement.pdf

[Figure]

Figure 2. Time profile of surface tension measurements in a non-logarithmic form.

**Fig. 1.** Figure 2. Time profile of surface tension measurements in a non-logarithmic form.

[Figure]

**Figure 4 (a).** Surface tension (σ) distribution of the 480 strains tested for biosurfactant production according to the physicochemical characteristics of cloud waters (marine, highly marine, continental and polluted). Highlighted in blue, the number of tested strains. Box and whisker plots are shown with the minimal (red) and maximal (green) surface tensions. The orange boxes represent 25th and 75th percentiles (lower and upper quartiles) of the measure **(b).** Phyla distribution according to the physicochemical characteristics of cloud waters (marine, highly marine, continental and polluted).

**Fig. 2.** Figure 4 (a). Surface tension ($\sigma$) distribution of the 480 strains tested for biosurfactant production according to the physicochemical characteristics of cloud waters (marine, highly marine, continent

**Supplement:**

**Institut de Chimie de Clermont-Ferrand (ICCF)**

[Figure]

Dr Anne-Marie DELORT
Directrice de recherche au CNRS

Aubière, The 24th of august 2016

To Prof. V. Faye McNeill
    Associate Editor, Atmospheric Chemistry and Physics

**New submission of manuscript acp-2016-447**

Dear editor,

Please find below our revised manuscript entitled" **Screening of cloud microorganisms isolated at the puy de Dôme (France) station for the production of biosurfactants",** by P. Renard, I. Canet, M. Sancelme, N. Wirgot, L. Deguillaume, and A.-M. Delort that we would like to publish in *Atmospheric Chemistry and Physics*; as well as, the supplementary material, the new figures 2 and 4ab, and the answers to the reviewers.

We would like to thank all the reviewers for their interest in our work and for all their remarks that greatly helped to improve the manuscript.

We have taken into account all the remarks of reviewers and changed the text accordingly. The corrections are underlined in yellow in the revised manuscript.

The main point was concerning the statistical analysis used to correlate the origin of air masses, considering their back trajectories and their chemical composition with the distribution of the microbial phyla in these air masses.

We agree with the referee that using back trajectories is not the best way to define the origin of clouds, so we have deleted all information related to this point (Figure 4, Figure 5a and Figure 6b were deleted). We only kept the categories defined from ACP analysis taking into account the physicochemical compositions of the cloud samples (highly marine, marine, continental, polluted) as described in Deguillaume et al (2014) (see new the Figure 4 and b, previously as Figure 5b and 6b).

Our statistical analyses are based on 39 cloud events sampled over more than 10 years with 480 different strains; this is the largest data set on cloud samples ever studied. However it is still difficult to make

✉ Chimie 4 - 24, avenue des Landais, BP 80026, 63171 AUBIERE Cedex – France
☎ : (33) 04 73 40 77.14  -  🖨 : (33) 04 73 40 77 17
✉ : A-Marie.Delort@univ-bpclermont.fr

[Figure]

[Figure]

[Figure]

**Institut de Chimie de Clermont-Ferrand (ICCF)**

[Figure]

statistics on samples with such intra- and inter-sample variations, therefore we have chosen to delete all the statistical data, rather keeping our results as simple observations.

We hope this revised paper will be publishable as an article in *Atmospheric Chemistry and Physics.*

Yours sincerely,

Dr Anne-Marie Delort

[Figure]

✉ Chimie 4 - 24, avenue des Landais, BP 80026, 63171 AUBIERE Cedex – France
☎ : (33) 04 73 40 77.14   -   🖷 : (33) 04 73 40 77 17
✍ : A-Marie.Delort@univ-bpclermont.fr

[revised manuscript text omitted]
 (Blue: marine, purple: highly marine, green: continental and black: polluted) as described by Deguillaume et al. (2014).

| Cloud Event | Composition | Nb of strains | Date | pH | SO$_4^{2-}$ | NO$_3^-$ | Cl$^-$ | Acetate | Formate | Oxalate | Succinate | Malonate | Na$^+$ | NH$_4^+$ | Mg$^{2+}$ | K$^+$ | Ca$^{2+}$ |
|---|---|---|---|---|---|---|---|---|---|---|---|---|---|---|---|---|---|
| 21 | Marine | 2 | 2004-01 | 5.6 | NA | NA | NA | NA | NA | NA | NA | NA | NA | NA | NA | NA | NA |
| 23 | Polluted | 2 | 2004-02 | 3.1 | NA | NA | NA | NA | NA | NA | NA | NA | NA | NA | NA | NA | NA |
| 29 | Marine | 1 | 2004-07 | 5.5 | NA | NA | NA | NA | NA | NA | NA | NA | NA | NA | NA | NA | NA |
| 30 | Marine | 3 | 2004-09 | 7.6 | 3.8 | 6.5 | 12.0 | 6.0 | 6.0 | 0.5 | 0.1 | 0.16 | 19.4 | 54.9 | 4.9 | 4.1 | 12.0 |
| 32 | Continental | 1 | 2004-12 | 5.5 | 72.1 | 95.8 | 31.5 | 0.0 | 0.3 | 0.3 | 0.1 | 0.17 | 74.4 | 132.4 | 7.6 | 9.0 | 73.7 |
| 42 | Continental | 25 | 2007-12 | 4.7 | 39.7 | 198.4 | 20.2 | 10.2 | 5.8 | 2.9 | 0.6 | 0.58 | 19.1 | 148.2 | 3.5 | 11.9 | 58.0 |
| 43 | Highly marine | 25 | 2008-01 | 5.9 | 9.4 | 21.4 | 81.4 | 11.4 | 6.7 | 1.2 | 0.2 | 0.28 | 315.7 | 35.9 | 11.8 | 13.7 | 26.0 |
| 44 | Continental | 14 | 2008-02 | 5.2 | 24.6 | 65.9 | 17.2 | 26.8 | 18.0 | 1.3 | 0.5 | 0.39 | 15.2 | 148.5 | 1.8 | 5.8 | 22.3 |
| 45 | Continental | 22 | 2008-04 | 4.6 | 44.7 | 65.5 | 31.7 | 17.7 | 17.9 | 2.8 | 0.9 | 0.89 | 33.3 | 122.5 | 4.9 | 9.7 | 53.6 |
| 46 | Continental | 2 | 2008-10 | 5.0 | 13.2 | 102.4 | 28.7 | 2.7 | 14.2 | 1.3 | 0.4 | 0.52 | 10.4 | 72.7 | 6.9 | 5.3 | 77.7 |
| 47 | Marine | 14 | 2008-11 | 5.4 | 6.5 | 33.2 | 15.7 | 5.3 | 8.0 | 0.8 | 0.2 | 0.33 | 8.2 | 13.7 | 1.4 | 6.3 | 14.4 |
| 49 | Marine | 6 | 2009-01 | 6.5 | 14.7 | 20.0 | 113.9 | 4.8 | 4.8 | 1.2 | 0.3 | 0.44 | 70.9 | 58.2 | 29.2 | 12.5 | 14.5 |
| 50 | Marine | 7 | 2009-02 | 4.9 | 7.1 | 23.3 | 72.5 | 22.2 | 13.4 | 0.8 | 0.2 | 0.56 | NA | NA | NA | NA | NA |
| 53 | Polluted | 8 | 2009-03 | 4.0 | 73.8 | 516.5 | 193.8 | 41.2 | 13.7 | 3.5 | 1.2 | 0.62 | 171.9 | 363.2 | 13.5 | 71.6 | 52.3 |
| 54 | Marine | 62 | 2009-11 | 5.2 | 2.3 | 13.3 | 30.5 | 4.4 | 15.0 | 1.7 | 0.1 | 0.34 | 37.0 | 6.6 | 12.1 | 20.5 | 0.0 |
| 55 | Marine | 11 | 2009-11 | 5.8 | 9.3 | 34.8 | 97.5 | 6.7 | 10.1 | 3.3 | 0.3 | 0.36 | 95.1 | 31.1 | 12.6 | 17.2 | 0.0 |
| 60 | Highly marine | 32 | 2010-03 | 5.5 | 39.0 | 9.7 | 231.5 | 3.5 | 4.3 | 1.2 | 0.0 | 0.00 | 114.1 | 28.6 | 12.9 | 12.8 | 2.7 |
| 61 | Marine | 2 | 2010-05 | 6.2 | 3.1 | 6.0 | 11.6 | 6.0 | 6.4 | 0.1 | 0.0 | 0.00 | 11.8 | 15.4 | 0.2 | 6.2 | 4.2 |
| 62 | Marine | 1 | 2010-06 | 6.1 | 3.5 | 4.5 | 2.3 | 3.4 | 6.2 | 1.4 | 0.0 | 0.00 | 1.8 | 6.0 | 0.0 | 0.0 | 0.0 |
| 66 | Marine | 1 | 2010-09 | 5.7 | 4.0 | 17.8 | 1.5 | 7.1 | 8.6 | 2.6 | 0.2 | 0.00 | 2.3 | 32.4 | 0.2 | 0.7 | 0.5 |
| 71 | Marine | 2 | 2011-03 | 5.9 | 3.5 | 6.6 | 6.2 | 7.1 | 4.2 | 2.2 | 0.3 | 0.39 | 6.9 | 42.5 | 0.1 | 3.6 | 5.1 |
| 72 | Marine | 5 | 2011-03 | 7.0 | 12.7 | 26.0 | 26.4 | 16.6 | 12.7 | 1.8 | 0.5 | 0.53 | 40.2 | 75.1 | 0.4 | 12.5 | 15.2 |
| 75 | Marine | 9 | 2011-06 | 5.2 | NA | NA | NA | NA | NA | NA | NA | NA | NA | NA | NA | NA | NA |
| 76 | Continental | 4 | 2011-06 | 5.9 | 52.2 | 126.0 | 16.9 | NA | NA | NA | NA | NA | NA | NA | NA | NA | NA |
| 77 | Continental | 11 | 2011-07 | 6.0 | 19.6 | 47.9 | 12.8 | 58.3 | 109.6 | 12.3 | NA | NA | 124.8 | 408.5 | 26.8 | 22.8 | 148 |
| 78 | Marine | 8 | 2011-07 | 5.5 | 7.1 | 10.4 | 24.1 | 3.3 | 11.3 | 3.1 | NA | NA | 22.7 | 32.7 | 12.3 | 9.2 | 7.1 |
| 79 | Marine | 5 | 2011-11 | 4.6 | 1.3 | 5.0 | 0.3 | 4.3 | 5.6 | 1.0 | NA | NA | 0.6 | 52.7 | 0.8 | 1.6 | 3.2 |
| 80 | Marine | 2 | 2012-01 | 4.9 | 1.8 | 2.2 | 13.4 | 7.4 | 5.3 | 1.1 | NA | NA | 80.6 | 39.3 | 12.0 | 6.1 | 5.2 |
| 81 | Marine | 7 | 2012-01 | 5.8 | 4.8 | 11.2 | 23.4 | 15.3 | 15.6 | 6.7 | NA | NA | 145.7 | 188.6 | 20.3 | 6.2 | 9.3 |
| 82 | Marine | 2 | 2012-03 | 5.3 | 1.6 | 2.0 | 0.2 | 4.2 | 6.2 | 3.0 | NA | NA | 1.0 | 75.4 | 1.0 | 0.1 | 6.1 |
| 83 | Continental | 10 | 2012-04 | 5.6 | 10.7 | 49.9 | 12.2 | 0.0 | 39.0 | 6.5 | NA | NA | 93.5 | 531.1 | 14.3 | 10.7 | 34.6 |
| 84 | Marine | 21 | 2012-04 | 5.5 | 1.6 | 1.9 | 6.0 | 3.9 | 7.7 | 1.2 | NA | NA | 36.0 | 38.1 | 6.0 | 1.4 | 5.2 |
| 85 | Marine | 17 | 2012-06 | 5.5 | 1.2 | 3.0 | 0.9 | 0.0 | 18.2 | 3.3 | NA | NA | 6.0 | 77.9 | 2.2 | 4.0 | 4.3 |
| 86 | Marine | 42 | 2012-09 | 5.9 | 0.5 | 1.0 | 1.0 | 3.2 | 3.2 | 1.0 | NA | NA | 8.8 | 16.8 | 1.4 | 5.5 | 4.5 |
| 87 | Marine | 28 | 2012-10 | 6.2 | 5.3 | 11.0 | 26.7 | 4.0 | 13.6 | 0.7 | NA | NA | 21.2 | 39.3 | 8.3 | 7.6 | 2.3 |
| 88 | Polluted | 2 | 2012-11 | 4.6 | 111.1 | 346.5 | 47.6 | 18.5 | 13.1 | 6.5 | NA | NA | 17.6 | 59.3 | 15.8 | 53.6 | 6.6 |
| 89 | Marine | 31 | 2013-01 | 5.2 | 32.9 | 29.5 | 109.9 | 19.9 | 15.6 | 2.7 | NA | NA | 77.5 | 81.2 | 21.5 | 4.8 | 23.5 |
| 91 | Marine | 7 | 2013-05 | 5.5 | 28.5 | 9.3 | 16.3 | 21.8 | 9.8 | 3.3 | NA | NA | 13.3 | 57.4 | 6.2 | 8.0 | 22.2 |
| 97 | Marine | 26 | 2014-02 | 5.8 | NA | NA | NA | NA | NA | NA | NA | NA | NA | NA | NA | NA | NA |

**Table S2.** 480 strains tested for biosurfactant production. Strains are isolated during 39 cloud events (from 2004 to 2014) gathered in four categories according to the physicochemical characteristics of the cloud waters (Blue: Marine, purple: Highly marine, green: Continental and black: Polluted, see table S1) as described by Deguillaume et al. (2014). Strains are differentiated into four main categories, according to the measured surface tension (red: σ ≤ 30, orange: 30 < σ ≤ 45, yellow: 45 < σ ≤ 55 and green: σ > 55 mN m$^{-1}$, see details in text). Phylum colors correspond to those of Figure 1. All surface tension measurements are performed using the pendant drop method with an OCA 15 Pro tensiometer (Data Physics, Germany). A.N, accession number in GenBank.

| Cloud Events | Composition | Strain | Phylum (class) | Species | AN | σ (mN m$^{-1}$) |
|---|---|---|---|---|---|---|
| 76 | Continental | 49b04 | ϒ-Proteobacteria | Pseudomonas sp. | KR922066 | 24.4 |
| 77 | Continental | 50b03 | ϒ-Proteobacteria | Pseudomonas sp. | KR922069 | 25.6 |
| 77 | Continental | 50b08 | ϒ-Proteobacteria | Pseudomonas sp. | KR922074 | 25.6 |
| 77 | Continental | 50b04 | ϒ-Proteobacteria | Pseudomonas sp. | KR922070 | 25.9 |
| 77 | Continental | 50b02 | ϒ-Proteobacteria | Pseudomonas sp. | KR922068 | 26.1 |
| 45 | Continental | 26b18 | Basidiomycota | Udeniomyces sp. | JF706566 | 33 |
| 44 | Continental | 25b01 | Basidiomycota | Bullera globispora | HQ260318 | 36 |
| 77 | Continental | 50b07 | ϒ-Proteobacteria | Pseudomonas syringae | KR922073 | 36 |
| 42 | Continental | 23b16 | Actinobacteria | Leifsonia sp. | HQ256777 | 39 |
| 77 | Continental | 50b09 | ϒ-Proteobacteria | Pseudomonas syringae | KR922075 | 39 |
| 44 | Continental | 25b04 | Basidiomycota | Udeniomyces sp. | HQ256877 | 39 |
| 76 | Continental | 49b03 | Unidentified yeast | unidentified | | 42 |
| 44 | Continental | 25b05 | ϒ-Proteobacteria | Pseudomonas sp. | HQ256806 | 43 |
| 45 | Continental | 26b25 | β-Proteobacteria | Variovorax sp. | HQ256810 | 44 |
| 44 | Continental | 25b07 | ϒ-Proteobacteria | Erwinia billingiae | HQ256807 | 46 |
| 44 | Continental | 25b11 | ϒ-Proteobacteria | Erwinia billingiae | HQ256802 | 46 |
| 44 | Continental | 25b13 | ϒ-Proteobacteria | Erwinia billingiae | HQ256804 | 46 |
| 77 | Continental | 50b05 | ϒ-Proteobacteria | Pseudomonas sp. | KR922071 | 48 |
| 42 | Continental | 23b26 | unidentified bacteria | unidentified | | 48 |
| 44 | Continental | 25b03 | Actinobacteria | Clavibacter michiganensis | HQ256805 | 49 |
| 45 | Continental | 26b30 | Actinobacteria | Clavibacter michiganensis | HQ256811 | 49 |
| 45 | Continental | 26b21 | Actinobacteria | Plantibacter sp. | HQ260322 | 49 |
| 42 | Continental | 23b05 | Actinobacteria | Rhodococcus sp. | HQ256785 | 49 |
| 42 | Continental | 23b27 | Actinobacteria | Rhodococcus sp. | HQ256783 | 49 |
| 83 | Continental | 56b23 | Basidiomycota | Udeniomyces sp. | | 49 |
| 42 | Continental | 23b28 | unidentified bacteria | unidentified | | 49 |
| 45 | Continental | 26b19 | Basidiomycota | Bullera globispora | JF706567 | 50 |
| 45 | Continental | 26b16 | α-Proteobacteria | Sphingomonas sp. | HQ256808 | 50 |
| 46 | Continental | 27b03 | ϒ-Proteobacteria | Pseudomonas sp. | HQ256813 | 51 |
| 77 | Continental | 50b10 | ϒ-Proteobacteria | Pseudomonas sp. | | 51 |
| 42 | Continental | 23b25 | Actinobacteria | Rhodococcus sp. | HQ256782 | 51 |
| 83 | Continental | 56b01 | Actinobacteria | Plantibacter sp. | | 52 |
| 83 | Continental | 56b13 | Actinobacteria | Rhodococcus sp. | | 52 |
| 44 | Continental | 25b09 | Basidiomycota | Udeniomyces sp. | HQ256880 | 52 |
| 83 | Continental | 56b08 | Actinobacteria | Clavibacter michiganensis | KR922100 | 53 |
| 83 | Continental | 56b03 | Actinobacteria | Rhodococcus sp. | KR922098 | 53 |
| 44 | Continental | 25b14 | Basidiomycota | Dioszegia fristingensis | | 54 |
| 42 | Continental | 23b15 | Actinobacteria | Microbacterium oxydans | HQ256776 | 54 |
| 42 | Continental | 23b20 | Actinobacteria | Microbacterium sp. | HQ256779 | 54 |
| 45 | Continental | 26b08 | Basidiomycota | Cryptococcus sp. | JF706563 | 55 |
| 77 | Continental | 50b06 | ϒ-Proteobacteria | Pseudomonas syringae | KR922072 | 55 |
| 42 | Continental | 23b22 | Basidiomycota | Cryptococcus victoriae | JF706548 | 56 |
| 44 | Continental | 25b06 | Basidiomycota | Dioszegia butyracea | HQ256878 | 56 |
| 83 | Continental | 56b25 | Actinobacteria | Frigoribacterium sp. | KR922104 | 56 |
| 45 | Continental | 26b32 | Ascomycota | Tetracladium sp. | JF706575 | 56 |
| 45 | Continental | 26b04 | Basidiomycota | Dioszegia sp. | JF706560 | 57 |
| 42 | Continental | 23b29 | Actinobacteria | Rhodococcus sp. | HQ256784 | 57 |
| 77 | Continental | 50b11 | α-Proteobacteria | Sphingomonas sp. | | 57 |
| 83 | Continental | 56b21 | α-Proteobacteria | Sphingomonas sp. | KR922103 | 57 |
| 45 | Continental | 26b23 | Actinobacteria | Subtercola sp. | HQ256809 | 57 |
| 45 | Continental | 26b11 | Basidiomycota | Cryptococcus sp. | JF706565 | 58 |
| 42 | Continental | 23b18 | Basidiomycota | Cryptococcus victoriae | JF706547 | 58 |
| 83 | Continental | 56b14 | Actinobacteria | Plantibacter sp. | KR922102 | 58 |
| 83 | Continental | 56b04 | α-Proteobacteria | Sphingomonas sp. | KR922099 | 58 |
| 42 | Continental | 23b21 | Actinobacteria | Subtercola boreus | HQ256780 | 58 |
| 45 | Continental | 26b06 | Basidiomycota | Dioszegia sp. | JF706562 | 59 |
| 45 | Continental | 26b34 | Basidiomycota | Dioszegia xingshenensis | JF706577 | 59 |
| 44 | Continental | 25b12 | Actinobacteria | Rhodococcus sp. | HQ256803 | 59 |
| 44 | Continental | 25b10 | Basidiomycota | Dioszegia sp. | HQ256875 | 60 |

| 42 | Continental | 23b24 | α-Proteobacteria | *Rhizobium* sp. | HQ256781 | 60 |
|---|---|---|---|---|---|---|
| 45 | Continental | 26b03 | Basidiomycota | *Cryptococcus* sp. | JF706559 | 61 |
| 42 | Continental | 23b07 | α-Proteobacteria | *Devosia* sp. | HQ256787 | 61 |
| 45 | Continental | 26b09 | Basidiomycota | *Dioszegia buhagiarii* | JF706564 | 61 |
| 45 | Continental | 26b20 | Basidiomycota | *Dioszegia* sp. | JF706568 | 62 |
| 45 | Continental | 26b24 | Basidiomycota | *Dioszegia* sp. | JF706570 | 62 |
| 45 | Continental | 26b26 | Basidiomycota | *Dioszegia* sp. | JF706571 | 62 |
| 45 | Continental | 26b05 | Basidiomycota | *Cryptococcus* sp. | JF706561 | 63 |
| 42 | Continental | 23b13 | Basidiomycota | *Dioszegia* sp. | JF706546 | 63 |
| 42 | Continental | 23b03 | Ascomycota | *Unidentified* | JF706545 | 63 |
| 44 | Continental | 25b02 | Basidiomycota | *Udeniomyces* sp. | HQ256876 | 64 |
| 44 | Continental | 25b08 | Basidiomycota | *Dioszegia butyracea* | HQ256879 | 65 |
| 77 | Continental | 50b01 | γ-Proteobacteria | *Pseudomonas grimondii* | KR922067 | 65 |
| 46 | Continental | 27b01 | α-Proteobacteria | *Sphingomonas* sp. | HQ256812 | 65 |
| 76 | Continental | 49b02 | α-Proteobacteria | *Sphingomonas* sp. | KR922065 | 65 |
| 76 | Continental | 49b01 | Unidentified yeast | *unidentified* | | 65 |
| 42 | Continental | 23b09 | Actinobacteria | *Streptomyces* sp. | HQ256788 | 66 |
| 42 | Continental | 23b14 | α-Proteobacteria | *Methylobacterium* sp. | HQ256775 | 67 |
| 42 | Continental | 23b19 | α-Proteobacteria | *Sphingomonas* sp. | HQ256787 | 67 |
| 83 | Continental | 56b24 | unidentified bacteria | *unidentified* | | 67 |
| 42 | Continental | 23b12 | Actinobacteria | *Streptomyces* sp. | | 68 |
| 42 | Continental | 23b02 | Ascomycota | *Unidentified* | JF706544 | 68 |
| 42 | Continental | 23b06 | α-Proteobacteria | *Sphingomonas* sp. | HQ256786 | 69 |
| 42 | Continental | 23b17 | Bacteroidetes | *Hymenobacter* sp. | HQ256778 | 70 |
| 45 | Continental | 26b27 | Basidiomycota | *Rhodotorula aurantiaca* | JF706572 | 70 |
| 42 | Continental | 23b11 | Actinobacteria | *Streptomyces* sp. | HQ256774 | 70 |
| 45 | Continental | 26b31 | Ascomycota | *Taphrina deformans* | JF706574 | 70 |
| 45 | Continental | 26b33 | Basidiomycota | *Dioszegia* sp. | JF706576 | 71 |
| 42 | Continental | 23b01 | α-Proteobacteria | *Sphingomonas* sp. | HQ256773 | 71 |
| 43 | Highly marine | 24b16 | Ascomycota | *Wickerhamomyces anomalus* | JF706554 | 45 |
| 43 | Highly marine | 24b04 | Actinobacteria | *Streptomyces microflavus* | HQ256797 | 47 |
| 43 | Highly marine | 24b12 | γ-Proteobacteria | *Pseudomonas* sp. | HQ260323 | 49 |
| 43 | Highly marine | 24b19 | Actinobacteria | *Clavibacter michiganensis* | HQ260320 | 50 |
| 43 | Highly marine | 24b26 | α-Proteobacteria | *Sphingomonas* sp. | | 50 |
| 43 | Highly marine | 24b10 | Actinobacteria | *Streptomyces microflavus* | HQ256790 | 50 |
| 43 | Highly marine | 24b13 | Ascomycota | *Wickerhamomyces anomalus* | JF706551 | 50 |
| 43 | Highly marine | 24b17 | Actinobacteria | *Clavibacter michiganensis* | HQ256792 | 51 |
| 43 | Highly marine | 24b24 | Actinobacteria | *Clavibacter michiganensis* | HQ256795 | 51 |
| 43 | Highly marine | 24b20 | Actinobacteria | *Frigoribacterium* sp. | HQ256793 | 51 |
| 43 | Highly marine | 24b18 | Basidiomycota | *Udeniomyces pannonicus* | JF706555 | 51 |
| 43 | Highly marine | 24b21 | Basidiomycota | *Udeniomyces pannonicus* | JF706556 | 51 |
| 43 | Highly marine | 24b05 | Actinobacteria | *Clavibacter michiganensis* | HQ256798 | 52 |
| 43 | Highly marine | 24b15 | Basidiomycota | *Dioszegia fristigensis* | JF706553 | 52 |
| 43 | Highly marine | 24b07 | Actinobacteria | *Curtobacterium flaccumfaciens* | HQ256800 | 53 |
| 43 | Highly marine | 24b23 | Actinobacteria | *Curtobacterium flaccumfaciens* | HQ256794 | 53 |
| 60 | Highly marine | 35b43 | Actinobacteria | *Rhodococcus* sp. | JF706519 | 53 |
| 60 | Highly marine | 35b14 | Firmicutes | *Bacillus* sp. | | 55 |
| 43 | Highly marine | 24b06 | Actinobacteria | *Curtobacterium flaccumfaciens* | HQ256799 | 55 |
| 43 | Highly marine | 24b09 | Actinobacteria | *Aeromicrobium* sp. | HQ256801 | 57 |
| 43 | Highly marine | 24b01 | Actinobacteria | *Clavibacter michiganensis* | HQ256789 | 58 |
| 60 | Highly marine | 35b15 | α-Proteobacteria | *Sphingomonas* sp. | JF706508 | 58 |
| 60 | Highly marine | 35b22 | unidentified bacteria | *unidentified* | | 58 |
| 60 | Highly marine | 35b13 | Actinobacteria | *Frigoribacterium* sp. | JF706507 | 59 |
| 43 | Highly marine | 24b08 | Basidiomycota | *Bullera armeniaca* | JF706550 | 60 |
| 60 | Highly marine | 35b18 | Actinobacteria | *Curtobacterium herbarum* | JF706509 | 60 |
| 60 | Highly marine | 35b40 | β-Proteobacteria | *Janthinobacterium* sp. | JF706518 | 60 |
| 43 | Highly marine | 24b02 | Basidiomycota | *Bullera armeniaca* | JF706549 | 61 |
| 60 | Highly marine | 35b26 | Basidiomycota | *Udeniomyces* sp. | JN176601 | 61 |
| 43 | Highly marine | 24b22 | Basidiomycota | *Bullera armeniaca* | JF706557 | 62 |
| 60 | Highly marine | 35b45 | Basidiomycota | *Dioszegia fristigensis* | JN176610 | 63 |
| 43 | Highly marine | 24b25 | Basidiomycota | *Udeniomyces* sp. | JF706558 | 63 |
| 60 | Highly marine | 35b29 | Basidiomycota | *Dioszegia butyracea* | JN176603 | 64 |
| 60 | Highly marine | 35b35 | Basidiomycota | *Dioszegia crocea* | JN176606 | 64 |
| 60 | Highly marine | 35b30 | Basidiomycota | *Dioszegia crocea* | JN176604 | 65 |
| 60 | Highly marine | 35b02 | α-Proteobacteria | *Sphingomonas* sp. | JF706510 | 65 |
| 60 | Highly marine | 35b39 | Basidiomycota | *Dioszegia crocea* | JN176607 | 66 |
| 60 | Highly marine | 35b42 | Basidiomycota | *Dioszegia fristingensis* | JN176608 | 66 |
| 60 | Highly marine | 35b01 | Actinobacteria | *Frigoribacterium* sp. | JF706506 | 66 |
| 60 | Highly marine | 35b17 | Basidiomycota | *Dioszegia crocea* | JN176597 | 67 |
| 60 | Highly marine | 35b44 | Basidiomycota | *Dioszegia crocea* | JN176609 | 67 |
| 43 | Highly marine | 24b14 | Basidiomycota | *Dioszegia* sp. | JF706552 | 67 |

| | | | | | | |
|---|---|---|---|---|---|---|
| 60 | Highly marine | 35b21 | Basidiomycota | *Mastigobasidium intermedium* | JN176599 | 67 |
| 60 | Highly marine | 35b23 | Basidiomycota | *Mastigobasidium intermedium* | JN176600 | 67 |
| 60 | Highly marine | 35b27 | α-Proteobacteria | *Methylobacterium* sp. | JF706512 | 67 |
| 60 | Highly marine | 35b20 | Actinobacteria | *Curtobacterium flaccumfaciens* | JF706511 | 68 |
| 60 | Highly marine | 35b33 | β-Proteobacteria | *Massilia* sp. | JF706514 | 68 |
| 60 | Highly marine | 35b38 | α-Proteobacteria | *Sphingomonas* sp. | JF706517 | 68 |
| 60 | Highly marine | 35b32 | α-Proteobacteria | *Sphingomonas* sp. | JF706513 | 69 |
| 60 | Highly marine | 35b12 | Basidiomycota | *Sporobolomyces roseus* | JF706594 | 69 |
| 43 | Highly marine | 24b03 | Actinobacteria | *Streptomyces* sp. | HQ256796 | 69 |
| 60 | Highly marine | 35b04 | Basidiomycota | *Cryptococcus* sp. | JN176596 | 70 |
| 60 | Highly marine | 35b19 | Basidiomycota | *Dioszegia crocea* | JN176598 | 70 |
| 60 | Highly marine | 35b28 | Basidiomycota | *Dioszegia fristigensis* | JN176602 | 70 |
| 60 | Highly marine | 35b34 | Actinobacteria | *Rhodococcus* sp. | JF706515 | 71 |
| 60 | Highly marine | 35b37 | Actinobacteria | *Rhodococcus* sp. | JF706516 | 71 |
| 60 | Highly marine | 35b31 | Basidiomycota | *Dioszegia crocea* | JN176605 | 72 |
| 78 | Marine | 51b07 | Υ-Proteobacteria | *Pseudomonas* sp. | KR922082 | 24.8 |
| 78 | Marine | 51b04 | Υ-Proteobacteria | *Pseudomonas* sp. | KR922079 | 25.2 |
| 87 | Marine | 60b24 | Υ-Proteobacteria | *Pseudomonas* sp. | KR922197 | 25.3 |
| 54 | Marine | 32b42 | Υ-Proteobacteria | *Pseudomonas* sp. | HQ256842 | 25.5 |
| 75 | Marine | 48b01 | Υ-Proteobacteria | *Pseudomonas* sp. | KR922057 | 25.7 |
| 78 | Marine | 51b06 | Υ-Proteobacteria | *Pseudomonas* sp. | KR922081 | 26 |
| 87 | Marine | 60b01 | Υ-Proteobacteria | *Pseudomonas* sp. | KR922180 | 26 |
| 85 | Marine | 58b28 | Υ-Proteobacteria | *Pseudomonas syringae* | KR922139 | 26 |
| 85 | Marine | 58b02 | Υ-Proteobacteria | *Pseudomonas syringae* | KR922125 | 26.4 |
| 86 | Marine | 59b12 | Υ-Proteobacteria | *Pseudomonas* sp. | KR922150 | 26.8 |
| 86 | Marine | 59b10 | Υ-Proteobacteria | *Pseudomonas* sp. | KR922148 | 27 |
| 54 | Marine | 32b53 | Υ-Proteobacteria | *Pseudomonas trivialis* | HQ256851 | 27 |
| 54 | Marine | 32b52 | Υ-Proteobacteria | *Xanthomonas campestris* | HQ256850 | 27 |
| 61 | Marine | 36b03 | Υ-Proteobacteria | *Pseudomonas fluorescens* | JF706525 | 27.5 |
| 86 | Marine | 59b11 | Υ-Proteobacteria | *Pseudomonas* sp. | KR922149 | 27.5 |
| 91 | Marine | 66b02 | Υ-Proteobacteria | *Pseudomonas syringae* | KR922247 | 27.5 |
| 75 | Marine | 48b02 | Υ-Proteobacteria | *Pseudomonas reinekei* | | 27.8 |
| 30 | Marine | 14b02 | Υ-Proteobacteria | *Pseudomonas* sp. | | 27.8 |
| 72 | Marine | 47b07 | Υ-Proteobacteria | *Pseudomonas* sp. | | 27.8 |
| 78 | Marine | 51b03 | Υ-Proteobacteria | *Pseudomonas* sp. | KR922078 | 28 |
| 54 | Marine | 32b32 | Υ-Proteobacteria | *Xanthomonas campestris* | JN176586 | 28.3 |
| 86 | Marine | 59b03 | Υ-Proteobacteria | *Pseudomonas syringae* | KR922141 | 28.4 |
| 54 | Marine | 32b22 | Υ-Proteobacteria | *Xanthomonas campestris* | JN176582 | 29 |
| 61 | Marine | 36b05 | Υ-Proteobacteria | *Pseudomonas fluorescens* | JF706526 | 29.1 |
| 54 | Marine | 32b74 | Υ-Proteobacteria | *Pseudomonas syringae* | HQ256872 | 29.1 |
| 84 | Marine | 57b01 | Υ-Proteobacteria | *Pseudomonas* sp. | KR922105 | 30 |
| 86 | Marine | 59b37 | Υ-Proteobacteria | *Pseudomonas* sp. | KR922166 | 30 |
| 55 | Marine | 33b02 | Υ-Proteobacteria | *Pseudomonas syringae* | HQ256867 | 30 |
| 54 | Marine | 32b24 | Υ-Proteobacteria | *Xanthomonas* sp. | JN176583 | 30 |
| 91 | Marine | 66b05 | Υ-Proteobacteria | *Pseudomonas graminis* | KR922249 | 33 |
| 78 | Marine | 51b05 | Υ-Proteobacteria | *Pseudomonas syringae* | KR922080 | 33 |
| 82 | Marine | 55b15 | Υ-Proteobacteria | *Pseudomonas syringae* | KR922097 | 34 |
| 86 | Marine | 59b32 | Υ-Proteobacteria | *Erwinia billingiae* | KR922162 | 35 |
| 75 | Marine | 48b05 | Υ-Proteobacteria | *Pseudomonas syringae* | KR922059 | 35 |
| 86 | Marine | 59b16 | Υ-Proteobacteria | *Pseudomonas syringae* | KR922153 | 36 |
| 54 | Marine | 32b27 | Basidiomycota | *Dioszegia xingshenensis* | JN176593 | 39 |
| 21 | Marine | 05b01 | Firmicutes | *Bacillus pumilus* | | 40 |
| 86 | Marine | 59b07 | Υ-Proteobacteria | *Pseudomonas* sp. | KR922145 | 40 |
| 86 | Marine | 59b25 | Υ-Proteobacteria | *Pseudomonas graminis* | KR922157 | 41 |
| 78 | Marine | 51b01 | Υ-Proteobacteria | *Pseudomonas* sp. | KR922077 | 41 |
| 54 | Marine | 32b67 | Υ-Proteobacteria | *Pseudomonas syringae* | HQ256864 | 42 |
| 47 | Marine | 28b11 | Basidiomycota | *Udeniomyces pannonicus* | HQ256882 | 42 |
| 86 | Marine | 59b41 | Υ-Proteobacteria | *Erwinia* sp. | KR922170 | 43 |
| 47 | Marine | 28b12 | Basidiomycota | *Udeniomyces* sp. | HQ256883 | 43 |
| 86 | Marine | 59b14 | Υ-Proteobacteria | *Pseudomonas syringae* | KR922151 | 44 |
| 47 | Marine | 28b02 | Basidiomycota | *Bannoa* sp. | JN176592 | 45 |
| 85 | Marine | 58b01 | Υ-Proteobacteria | *Pseudomonas syringae* | KR922124 | 45 |
| 91 | Marine | 66b14 | α-Proteobacteria | *Sphingomonas* sp. | KR922251 | 45 |
| 84 | Marine | 57b26 | Actinobacteria | *Clavibacter michiganensis* | KR922121 | 46 |
| 30 | Marine | 14b13 | Actinobacteria | *Frigoribacterium* sp. | DQ512796 | 46 |
| 86 | Marine | 59b02 | Υ-Proteobacteria | *Pseudomonas syringae* | | 46 |
| 85 | Marine | 58b25 | Υ-Proteobacteria | *Pseudomonas* sp. | KR922137 | 47 |
| 86 | Marine | 59b04 | Υ-Proteobacteria | *Pseudomonas* sp. | KR922142 | 47 |
| 78 | Marine | 51b10 | Υ-Proteobacteria | *Pseudomonas syringae* | KR922084 | 47 |
| 86 | Marine | 59b05 | Υ-Proteobacteria | *Pseudomonas syringae* | KR922143 | 47 |
| 49 | Marine | 29b06 | Basidiomycota | *Udeniomyces pannonicus* | HQ256895 | 47 |

| | | | | | | |
|---|---|---|---|---|---|---|
| 54 | Marine | 32b64 | Basidiomycota | *Udeniomyces* sp. | JF706586 | 47 |
| 84 | Marine | 57b22 | Actinobacteria | *Clavibacter michiganensis* | KR922119 | 48 |
| 86 | Marine | 59b58 | Actinobacteria | *Clavibacter michiganensis* | KR922179 | 48 |
| 54 | Marine | 32b56 | Bacteroidetes | *Flavobacterium* sp. | HQ256854 | 48 |
| 29 | Marine | 13b03 | Υ-Proteobacteria | *Pseudomonas graminis* | DQ512786 | 48 |
| 54 | Marine | 32b55 | Υ-Proteobacteria | *Pseudomonas graminis* | HQ256853 | 48 |
| 50 | Marine | 30b02 | Actinobacteria | *Rhodococcus* sp. | HQ256817 | 48 |
| 50 | Marine | 30b05 | Actinobacteria | *Rhodococcus* sp. | HQ256819 | 48 |
| 84 | Marine | 57b11 | α-Proteobacteria | *Sphingomonas* sp. | KR922112 | 48 |
| 49 | Marine | 29b04 | Basidiomycota | *Sporobolomyces roseus* | HQ256893 | 48 |
| 49 | Marine | 29b05 | Basidiomycota | *Udeniomyces pannonicus* | HQ256894 | 48 |
| 21 | Marine | 05b02 | Firmicutes | *Bacillus* sp. | | 49 |
| 47 | Marine | 28b04 | Basidiomycota | *Bensingtonia yucciola* | HQ256887 | 49 |
| 54 | Marine | 32b09 | Basidiomycota | *Dioszegia hungarica* | HQ256898 | 49 |
| 75 | Marine | 48b10 | Υ-Proteobacteria | *Pseudomonas* sp. | KR922063 | 49 |
| 86 | Marine | 59b15 | Υ-Proteobacteria | *Pseudomonas syringae* | KR922152 | 49 |
| 84 | Marine | 57b16 | Actinobacteria | *Clavibacter michiganensis* | KR922117 | 50 |
| 86 | Marine | 59b50 | Actinobacteria | *Clavibacter* sp. | KR922173 | 50 |
| 87 | Marine | 60b04 | Actinobacteria | *Frigoribacterium* sp. | KR922183 | 50 |
| 85 | Marine | 58b20 | Υ-Proteobacteria | *Pseudomonas graminis* | KR922133 | 50 |
| 86 | Marine | 59b01 | Υ-Proteobacteria | *Pseudomonas* sp. | KR922140 | 50 |
| 86 | Marine | 59b06 | Υ-Proteobacteria | *Pseudomonas* sp. | KR922144 | 50 |
| 86 | Marine | 59b17 | Υ-Proteobacteria | *Pseudomonas* sp. | KR922154 | 50 |
| 86 | Marine | 59b40 | Υ-Proteobacteria | *Pseudomonas* sp. | KR922169 | 50 |
| 89 | Marine | 63b11 | Actinobacteria | *Clavibacter* sp. | KR922218 | 51 |
| 85 | Marine | 58b05 | Actinobacteria | *Curtobacterium* sp. | KR922127 | 51 |
| 86 | Marine | 59b26 | Υ-Proteobacteria | *Dyella* sp. | KR922158 | 51 |
| 75 | Marine | 48b07 | Actinobacteria | *Frigoribacterium* sp. | KR922060 | 51 |
| 91 | Marine | 66b01 | Υ-Proteobacteria | *Pseudomonas syringae* | KR922246 | 51 |
| 84 | Marine | 57b28 | α-Proteobacteria | *Sphingomonas* sp. | KR922123 | 51 |
| 47 | Marine | 28b10 | Basidiomycota | *Udeniomyces pannonicus* | HQ256881 | 51 |
| 84 | Marine | 57b10 | Firmicutes | *Bacillus* sp. | KR922111 | 52 |
| 86 | Marine | 59b30 | Actinobacteria | *Clavibacter michiganensis* | KR922161 | 52 |
| 54 | Marine | 32b35 | Actinobacteria | *Frigoribacterium* sp. | HQ256839 | 52 |
| 86 | Marine | 59b34 | Actinobacteria | *Frigoribacterium* sp. | KR922164 | 52 |
| 86 | Marine | 59b57 | Bacteroidetes | *Pedobacter agri* | KR922178 | 52 |
| 85 | Marine | 58b21 | Bacteroidetes | *Pedobacter* sp. | KR922134 | 52 |
| 91 | Marine | 66b03 | Bacteroidetes | *Pedobacter* sp. | KR922248 | 52 |
| 86 | Marine | 59b38 | Υ-Proteobacteria | *Pseudomonas rhizosphaerae* | KR922167 | 52 |
| 86 | Marine | 59b29 | Actinobacteria | *Rhodococcus* sp. | | 52 |
| 84 | Marine | 57b13 | α-Proteobacteria | *Sphingomonas* sp. | KR922114 | 52 |
| 84 | Marine | 57b07 | unidentified bacteria | *unidentified* | | 52 |
| 89 | Marine | 63b09 | Actinobacteria | *Clavibacter michiganensis* | KR922216 | 53 |
| 84 | Marine | 57b12 | Actinobacteria | *Curtobacterium* sp. | KR922113 | 53 |
| 55 | Marine | 33b11 | Basidiomycota | *Dioszegia hungarica* | JF706591 | 53 |
| 50 | Marine | 30b01 | Actinobacteria | *Frigoribacterium* sp. | HQ256816 | 53 |
| 82 | Marine | 55b02 | Υ-Proteobacteria | *Pseudomonas* sp. | KR922096 | 53 |
| 86 | Marine | 59b53 | Υ-Proteobacteria | *Pseudomonas* sp. | KR922175 | 53 |
| 47 | Marine | 28b15 | Basidiomycota | *Rhodotorula* sp. | HQ256885 | 53 |
| 55 | Marine | 33b12 | α-Proteobacteria | *Sphingomonas* sp. | HQ256874 | 53 |
| 87 | Marine | 60b31 | α-Proteobacteria | *Sphingomonas* sp. | KR922202 | 53 |
| 97 | Marine | 67b09 | α-Proteobacteria | *Sphingomonas* sp. | KR922258 | 53 |
| 97 | Marine | 67b28 | Actinobacteria | *Clavibacter michiganensis* | KR922268 | 54 |
| 47 | Marine | 28b03 | Basidiomycota | *Cryptococcus victoriae* | HQ256886 | 54 |
| 85 | Marine | 58b17 | Actinobacteria | *Curtobacterium* sp. | | 54 |
| 86 | Marine | 59b28 | Υ-Proteobacteria | *Dyella* sp. | KR922160 | 54 |
| 85 | Marine | 58b04 | Actinobacteria | *Frigoribacterium* sp. | KR922126 | 54 |
| 89 | Marine | 63b02 | Actinobacteria | *Frigoribacterium* sp. | KR922209 | 54 |
| 86 | Marine | 59b08 | Υ-Proteobacteria | *Pseudomonas fluorescens* | KR922146 | 54 |
| 54 | Marine | 32b66 | Υ-Proteobacteria | *Pseudomonas graminis* | HQ256863 | 54 |
| 75 | Marine | 48b11 | α-Proteobacteria | *Sphingomonas* sp. | KR922064 | 54 |
| 81 | Marine | 54b07 | α-Proteobacteria | *Sphingomonas* sp. | KR922094 | 54 |
| 87 | Marine | 60b16 | α-Proteobacteria | *Sphingomonas* sp. | KR922193 | 54 |
| 87 | Marine | 60b22 | α-Proteobacteria | *Sphingomonas* sp. | KR922195 | 54 |
| 97 | Marine | 67b22 | Basidiomycota | *Udeniomyces pannonicus* | KR922311 | 54 |
| 54 | Marine | 32b05 | Basidiomycota | *Udeniomyces* sp. | HQ256897 | 54 |
| 49 | Marine | 29b03 | Firmicutes | *Bacillus* sp. | JF706502 | 55 |
| 54 | Marine | 32b13 | Actinobacteria | *Clavibacter* sp. | HQ256832 | 55 |
| 97 | Marine | 67b30 | Actinobacteria | *Clavibacter* sp. | KR922269 | 55 |
| 86 | Marine | 59b27 | Υ-Proteobacteria | *Dyella* sp. | KR922159 | 55 |
| 75 | Marine | 48b09 | Υ-Proteobacteria | *Pantoe* sp. | KR922062 | 55 |

| | | | | | |
|---|---|---|---|---|---|
| 87 | Marine | 60b12 | Ɣ-Proteobacteria | *Pseudomonas* sp. | KR922190 | 55 |
| 89 | Marine | 63b30 | Ɣ-Proteobacteria | *Pseudomonas* sp. | KR922230 | 55 |
| 49 | Marine | 29b02 | Actinobacteria | *Rhodococcus* sp. | HQ256815 | 55 |
| 87 | Marine | 60b23 | α-Proteobacteria | *Sphingomonas* sp. | KR922196 | 55 |
| 54 | Marine | 32b21 | Basidiomycota | *Udeniomyces* sp. | JF706580 | 55 |
| 97 | Marine | 67b25 | Basidiomycota | *Udeniomyces* sp. | KR922312 | 55 |
| 49 | Marine | 29b01 | Basidiomycota | *Bensingtonia yucciola* | HQ256892 | 56 |
| 87 | Marine | 60b03 | α-Proteobacteria | *Brevundimonas* sp. | KR922182 | 56 |
| 71 | Marine | 46b09 | Actinobacteria | *Curtobacterium* sp. | | 56 |
| 54 | Marine | 32b58 | Ɣ-Proteobacteria | *Ewingella americana* | JF706505 | 56 |
| 84 | Marine | 57b14 | Actinobacteria | *Frigoribacterium* sp. | KR922115 | 56 |
| 86 | Marine | 59b54 | Actinobacteria | *Frigoribacterium* sp. | KR922176 | 56 |
| 86 | Marine | 59b44 | Ɣ-Proteobacteria | *Pantoe agglomerans* | KR922172 | 56 |
| 54 | Marine | 32b60 | Ɣ-Proteobacteria | *Pseudomonas graminis* | HQ256858 | 56 |
| 86 | Marine | 59b42 | α-Proteobacteria | *Sphingomonas* sp. | KR922171 | 56 |
| 54 | Marine | 32b12 | Basidiomycota | *Udeniomyces pannonicus* | JF706578 | 56 |
| 85 | Marine | 58b23 | Bacteroidetes | *Epiltholimonas* sp. | | 57 |
| 89 | Marine | 63b39 | Bacteroidetes | *Pedobacter* sp. | KR922237 | 57 |
| 66 | Marine | 41b02 | Actinobacteria | *Rhodococcus* sp. | JF706542 | 57 |
| 97 | Marine | 67b23 | α-Proteobacteria | *Sphingomonas* sp. | KR922265 | 57 |
| 89 | Marine | 63b42 | Actinobacteria | *Subtercola* sp. | KR922240 | 57 |
| 54 | Marine | 32b02 | Actinobacteria | *Agreia* sp. | HQ256834 | 58 |
| 71 | Marine | 46b08 | Actinobacteria | *Curtobacterium* sp. | KR922054 | 58 |
| 89 | Marine | 63b08 | Actinobacteria | *Frondihabitans* sp. | KR922215 | 58 |
| 85 | Marine | 58b22 | Bacteroidetes | *Pedobacter* sp. | KR922135 | 58 |
| 50 | Marine | 30b06 | Actinobacteria | *Rhodococcus* sp. | HQ256820 | 58 |
| 79 | Marine | 52b02 | Actinobacteria | *Rhodococcus* sp. | KR922085 | 58 |
| 87 | Marine | 60b06 | α-Proteobacteria | *Sphingomonas* sp. | KR922184 | 58 |
| 87 | Marine | 60b11 | α-Proteobacteria | *Sphingomonas* sp. | KR922189 | 58 |
| 89 | Marine | 63b43 | α-Proteobacteria | *Sphingomonas* sp. | KR922241 | 58 |
| 85 | Marine | 58b06 | unidentified bacteria | *unidentified* | | 58 |
| 87 | Marine | 60b10 | Ɣ-Proteobacteria | *Xanthomonas* sp. | KR922188 | 58 |
| 50 | Marine | 30b04 | Actinobacteria | *Agreia* sp. | HQ256818 | 59 |
| 84 | Marine | 57b27 | Actinobacteria | *Clavibacter michiganensis* | KR922122 | 59 |
| 89 | Marine | 63b38 | Actinobacteria | *Clavibacter michiganensis* | KR922236 | 59 |
| 84 | Marine | 57b03 | Actinobacteria | *Clavibacter* sp. | KR922107 | 59 |
| 97 | Marine | 67b17 | Actinobacteria | *Nocardioides* sp. | | 59 |
| 97 | Marine | 67b19 | Ɣ-Proteobacteria | *Pseudomonas* sp. | | 59 |
| 97 | Marine | 67b29 | Actinobacteria | *Rhodococcus* sp. | | 59 |
| 86 | Marine | 59b39 | α-Proteobacteria | *Sphingomonas* sp. | KR922168 | 59 |
| 87 | Marine | 60b32 | α-Proteobacteria | *Sphingomonas* sp. | KR922203 | 59 |
| 54 | Marine | 32b68 | Basidiomycota | *Udeniomyces* sp. | JF706587 | 59 |
| 97 | Marine | 67b04 | Firmicutes | *Bacillus* sp. | KR922255 | 60 |
| 86 | Marine | 59b20 | α-Proteobacteria | *Brevundimonas* sp. | KR922156 | 60 |
| 87 | Marine | 60b15 | α-Proteobacteria | *Brevundimonas* sp. | | 60 |
| 47 | Marine | 28b14 | β-Proteobacteria | *Burkholderia* sp. | HQ256814 | 60 |
| 81 | Marine | 54b02 | Actinobacteria | *Clavibacter michiganensis* | KR922091 | 60 |
| 47 | Marine | 28b07 | Basidiomycota | *Cryptococcus* sp. | HQ256890 | 60 |
| 89 | Marine | 63b03 | Actinobacteria | *Frigoribacterium* sp. | KR922210 | 60 |
| 54 | Marine | 32b61 | α-Proteobacteria | *Sphingomonas* sp. | HQ256859 | 60 |
| 89 | Marine | 63b40 | α-Proteobacteria | *Sphingomonas* sp. | KR922238 | 60 |
| 54 | Marine | 32b40 | Basidiomycota | *Udeniomyces* sp. | JF706582 | 60 |
| 86 | Marine | 59b33 | Firmicutes | *Bacillus* sp. | KR922163 | 61 |
| 47 | Marine | 28b13 | Basidiomycota | *Cryptococcus* sp. | HQ256884 | 61 |
| 85 | Marine | 58b16 | Actinobacteria | *Curtobacterium* sp. | KR922130 | 61 |
| 54 | Marine | 32b59 | Ɣ-Proteobacteria | *Ewingella americana* | HQ256856 | 61 |
| 54 | Marine | 32b63 | Actinobacteria | *Plantibacter* sp. | HQ256861 | 61 |
| 54 | Marine | 32b03 | Actinobacteria | *Rathayibacter* sp. | JF706503 | 61 |
| 87 | Marine | 60b07 | α-Proteobacteria | *Sphingomonas* sp. | KR922185 | 61 |
| 87 | Marine | 60b30 | α-Proteobacteria | *Sphingomonas* sp. | KR922201 | 61 |
| 89 | Marine | 63b41 | Actinobacteria | *Subtercola* sp. | KR922239 | 61 |
| 54 | Marine | 32b65 | α-Proteobacteria | *Agrobacterium* sp. | HQ256862 | 62 |
| 97 | Marine | 67b21 | Basidiomycota | *Cryptococcus* sp. | KR922310 | 62 |
| 54 | Marine | 32b36 | Actinobacteria | *Curtobacterium* sp. | HQ256840 | 62 |
| 47 | Marine | 28b05 | Basidiomycota | *Dioszegia crocea* | HQ256888 | 62 |
| 54 | Marine | 32b33 | Ɣ-Proteobacteria | *Pseudoxanthomonas* sp. | HQ256838 | 62 |
| 54 | Marine | 32b45 | α-Proteobacteria | *Sphingomonas* sp. | HQ256844 | 62 |
| 54 | Marine | 32b50 | α-Proteobacteria | *Sphingomonas* sp. | HQ256848 | 62 |
| 55 | Marine | 33b06 | Basidiomycota | *Udeniomyces* sp. | JF706589 | 62 |
| 86 | Marine | 59b55 | β-Proteobacteria | *Variovorax paradoxus* | KR922177 | 62 |
| 89 | Marine | 63b05 | Actinobacteria | *Arthrobacter nicotinovorans* | KR922212 | 63 |

| | | | | | | |
|---|---|---|---|---|---|---|
| 97 | Marine | 67b08 | Basidiomycota | *Cryptococcus* sp. | KR922308 | 63 |
| 55 | Marine | 33b07 | Basidiomycota | *Cryptococcus victoriae* | JF706590 | 63 |
| 86 | Marine | 59b36 | Actinobacteria | *Curtobacterium* sp. | KR922165 | 63 |
| 87 | Marine | 60b28 | Basidiomycota | *Dioszegia zslotii* | | 63 |
| 89 | Marine | 63b19 | Actinobacteria | *Frigoribacterium* sp. | KR922223 | 63 |
| 54 | Marine | 32b31 | γ-Proteobacteria | *Pseudomonas syringae* | JN176585 | 63 |
| 86 | Marine | 59b51 | Actinobacteria | *Rathayibacter* sp. | KR922174 | 63 |
| 87 | Marine | 60b29 | α-Proteobacteria | *Sphingomonas* sp. | KR922200 | 63 |
| 89 | Marine | 63b15 | α-Proteobacteria | *Sphingomonas* sp. | KR922221 | 63 |
| 91 | Marine | 66b11 | α-Proteobacteria | *Sphingomonas* sp. | KR922250 | 63 |
| 97 | Marine | 67b20 | Basidiomycota | *Udeniomyces pannonicus* | KR922309 | 63 |
| 85 | Marine | 58b15 | β-Proteobacteria | *Variovorax* sp. | KR922129 | 63 |
| 75 | Marine | 48b04 | Bacteroidetes | *Chryseobacterium* sp. | KR922058 | 64 |
| 89 | Marine | 63b20 | Actinobacteria | *Clavibacter michiganensis* | KR922224 | 64 |
| 89 | Marine | 63b47 | Actinobacteria | *Clavibacter michiganensis* | KR922244 | 64 |
| 47 | Marine | 28b06 | Basidiomycota | *Dioszegia crocea* | HQ256889 | 64 |
| 54 | Marine | 32b15 | Basidiomycota | *Dioszegia* sp. | JF706579 | 64 |
| 54 | Marine | 32b17 | Actinobacteria | *Frigoribacterium* sp. | HQ256833 | 64 |
| 50 | Marine | 30b07 | γ-Proteobacteria | *Psychrobacter* sp. | HQ256821 | 64 |
| 54 | Marine | 32b29 | α-Proteobacteria | *Rhizobium* sp. | HQ256837 | 64 |
| 54 | Marine | 32b11 | α-Proteobacteria | *Sphingomonas* sp. | HQ256831 | 64 |
| 87 | Marine | 60b13 | α-Proteobacteria | *Sphingomonas* sp. | KR922191 | 64 |
| 89 | Marine | 63b10 | α-Proteobacteria | *Sphingomonas* sp. | KR922217 | 64 |
| 89 | Marine | 63b27 | α-Proteobacteria | *Sphingomonas* sp. | KR922229 | 64 |
| 89 | Marine | 63b36 | α-Proteobacteria | *Sphingomonas* sp. | KR922234 | 64 |
| 54 | Marine | 32b14 | unidentified bacteria | *unidentified* | | 64 |
| 89 | Marine | 63b21 | β-Proteobacteria | *Variovorax* sp. | KR922225 | 64 |
| 32 | Marine | 16b01 | Firmicutes | *Bacillus* sp. | | 65 |
| 97 | Marine | 67b11 | α-Proteobacteria | *Brevundimonas* sp. | KR922259 | 65 |
| 89 | Marine | 63b34 | Bacteroidetes | *Chryseobacterium* sp. | | 65 |
| 47 | Marine | 28b01 | Basidiomycota | *Dioszegia crocea* | HQ260319 | 65 |
| 54 | Marine | 32b41 | Basidiomycota | *Dioszegia* sp. | JF706583 | 65 |
| 54 | Marine | 32b28 | Actinobacteria | *Frondihabitans* sp. | HQ256836 | 65 |
| 80 | Marine | 53b01 | γ-Proteobacteria | *Moraxella* sp. | KR922089 | 65 |
| 97 | Marine | 67b16 | Actinobacteria | *Promicromonospora* sp. | KR922263 | 65 |
| 84 | Marine | 57b25 | α-Proteobacteria | *Sphingomonas* sp. | KR922120 | 65 |
| 86 | Marine | 59b19 | α-Proteobacteria | *Sphingomonas* sp. | KR922155 | 65 |
| 54 | Marine | 32b51 | Actinobacteria | *Subtercola* sp. | HQ256849 | 65 |
| 84 | Marine | 57b19 | unidentified bacteria | *unidentified* | | 65 |
| 84 | Marine | 57b06 | Actinobacteria | *Arthrobacter* sp. | KR922109 | 66 |
| 54 | Marine | 32b46 | Actinobacteria | *Curtobacterium herbarum* | HQ256845 | 66 |
| 54 | Marine | 32b73 | Basidiomycota | *Dioszegia fristigensis* | JN176595 | 66 |
| 54 | Marine | 32b37 | Basidiomycota | *Dioszegia* sp. | JF706581 | 66 |
| 87 | Marine | 60b27 | β-Proteobacteria | *Duganella* sp. | KR922199 | 66 |
| 85 | Marine | 58b27 | Bacteroidetes | *Flavobacterium* sp. | KR922138 | 66 |
| 86 | Marine | 59b35 | Bacteroidetes | *Hymenobacter* sp. | | 66 |
| 89 | Marine | 63b13 | β-Proteobacteria | *Janthinobacterium* sp. | KR922220 | 66 |
| 54 | Marine | 32b20 | Actinobacteria | *Leifsonia* sp. | HQ256835 | 66 |
| 50 | Marine | 30b03 | Actinobacteria | *Nocardioides* sp. | | 66 |
| 81 | Marine | 54b03 | Actinobacteria | *Rhodococcus* sp. | KR922092 | 66 |
| 30 | Marine | 14b06 | α-Proteobacteria | *Sphingomonas* sp. | DQ512790 | 66 |
| 84 | Marine | 57b15 | α-Proteobacteria | *Sphingomonas* sp. | KR922116 | 66 |
| 97 | Marine | 67b13 | Actinobacteria | *Streptomyces* sp. | KR922260 | 66 |
| 54 | Marine | 32b44 | Basidiomycota | *Dioszegia hungarica* | JF706584 | 67 |
| 85 | Marine | 58b07 | Bacteroidetes | *Epiltholimonas* sp. | | 67 |
| 97 | Marine | 67b15 | Actinobacteria | *Janibacter* sp. | KR922262 | 67 |
| 54 | Marine | 32b54 | Actinobacteria | *Rathayibacter* sp. | HQ256852 | 67 |
| 97 | Marine | 67b07 | α-Proteobacteria | *Rhizobium* sp. | KR922257 | 67 |
| 84 | Marine | 57b04 | α-Proteobacteria | *Sphingomonas* sp. | KR922108 | 67 |
| 87 | Marine | 60b20 | α-Proteobacteria | *Sphingomonas* sp. | | 67 |
| 89 | Marine | 63b35 | α-Proteobacteria | *Sphingomonas* sp. | KR922233 | 67 |
| 89 | Marine | 63b45 | α-Proteobacteria | *Sphingomonas* sp. | KR922242 | 67 |
| 97 | Marine | 67b24 | α-Proteobacteria | *Sphingomonas* sp. | KR922266 | 67 |
| 54 | Marine | 32b71 | Basidiomycota | *Udeniomyces puniceus* | JN176594 | 67 |
| 97 | Marine | 67b32 | Unidentified yeast | *unidentified* | | 67 |
| 72 | Marine | 47b02 | β-Proteobacteria | *Variovorax* sp. | KR922055 | 67 |
| 54 | Marine | 32b48 | Basidiomycota | *Bullera globispora* | JF706585 | 68 |
| 85 | Marine | 58b19 | Bacteroidetes | *Hymenobacter* sp. | KR922132 | 68 |
| 72 | Marine | 47b01 | α-Proteobacteria | *Sphingomonas* sp. | | 68 |
| 79 | Marine | 52b08 | α-Proteobacteria | *Sphingomonas* sp. | KR922088 | 68 |
| 87 | Marine | 60b08 | α-Proteobacteria | *Sphingomonas* sp. | KR922186 | 68 |

| | | | | | | |
|---|---|---|---|---|---|---|
| 87 | Marine | 60b09 | α-Proteobacteria | *Sphingomonas* sp. | KR922187 | 68 |
| 89 | Marine | 63b04 | α-Proteobacteria | *Sphingomonas* sp. | KR922211 | 68 |
| 89 | Marine | 63b37 | α-Proteobacteria | *Sphingomonas* sp. | KR922235 | 68 |
| 54 | Marine | 32b47 | γ-Proteobacteria | *Stenotrophomonas* sp. | HQ256846 | 68 |
| 55 | Marine | 33b05 | γ-Proteobacteria | *Stenotrophomonas* sp. | HQ256869 | 68 |
| 55 | Marine | 33b04 | Unidentified yeast | *unidentified* | | 68 |
| 97 | Marine | 67b27 | Basidiomycota | *Cryptococcus aquaticus* | KR922313 | 69 |
| 72 | Marine | 47b05 | Bacteroidetes | *Hymenobacter* sp. | KR922056 | 69 |
| 91 | Marine | 67b01 | β-Proteobacteria | *Janthinobacterium* sp. | KR922252 | 69 |
| 62 | Marine | 37b05 | γ-Proteobacteria | *Pseudomonas fluorescens* | JF706535 | 69 |
| 78 | Marine | 51b08 | γ-Proteobacteria | *Pseudomonas* sp. | KR922083 | 69 |
| 81 | Marine | 54b04 | α-Proteobacteria | *Sphingomonas* sp. | | 69 |
| 81 | Marine | 54b06 | α-Proteobacteria | *Sphingomonas* sp. | KR922093 | 69 |
| 84 | Marine | 57b02 | α-Proteobacteria | *Sphingomonas* sp. | KR922106 | 69 |
| 84 | Marine | 57b08 | α-Proteobacteria | *Sphingomonas* sp. | KR922110 | 69 |
| 89 | Marine | 63b07 | α-Proteobacteria | *Sphingomonas* sp. | KR922214 | 69 |
| 89 | Marine | 63b12 | α-Proteobacteria | *Sphingomonas* sp. | KR922219 | 69 |
| 89 | Marine | 63b26 | α-Proteobacteria | *Sphingomonas* sp. | KR922228 | 69 |
| 89 | Marine | 63b33 | α-Proteobacteria | *Sphingomonas* sp. | KR922232 | 69 |
| 97 | Marine | 67b03 | α-Proteobacteria | *Sphingomonas* sp. | KR922254 | 69 |
| 97 | Marine | 67b14 | Firmicutes | *Staphylococcus xylosus* | KR922261 | 69 |
| 79 | Marine | 52b05 | γ-Proteobacteria | *Stenotrophomonas* sp. | KR922087 | 69 |
| 97 | Marine | 67b06 | Actinobacteria | *Cellulomonas cellasea* | KR922256 | 70 |
| 87 | Marine | 60b25 | β-Proteobacteria | *Massilia* sp. | | 70 |
| 80 | Marine | 53b06 | Actinobacteria | *Nocardioides* sp. | KR922090 | 70 |
| 54 | Marine | 32b49 | α-Proteobacteria | *Sphingomonas* sp. | HQ256847 | 70 |
| 75 | Marine | 48b08 | α-Proteobacteria | *Sphingomonas* sp. | KR922061 | 70 |
| 79 | Marine | 52b03 | α-Proteobacteria | *Sphingomonas* sp. | KR922086 | 70 |
| 81 | Marine | 54b05 | α-Proteobacteria | *Sphingomonas* sp. | | 70 |
| 87 | Marine | 60b19 | α-Proteobacteria | *Sphingomonas* sp. | KR922194 | 70 |
| 97 | Marine | 67b26 | α-Proteobacteria | *Sphingomonas* sp. | KR922267 | 70 |
| 54 | Marine | 32b70 | Basidiomycota | *Sporobolomyces ruberrimus* | JF706588 | 70 |
| 54 | Marine | 32b43 | γ-Proteobacteria | *Stenotrophomonas* sp. | HQ256843 | 70 |
| 54 | Marine | 32b72 | γ-Proteobacteria | *Stenotrophomonas* sp. | HQ256871 | 70 |
| 55 | Marine | 33b03 | γ-Proteobacteria | *Xanthomonas campestris* | HQ256868 | 70 |
| 87 | Marine | 60b02 | γ-Proteobacteria | *Xanthomonas* sp. | KR922181 | 70 |
| 54 | Marine | 32b06 | Bacteroidetes | *Flavobacterium* sp. | HQ256857 | 71 |
| 81 | Marine | 54b11 | Bacteroidetes | *Hymenobacter* sp. | KR922095 | 71 |
| 54 | Marine | 32b57 | α-Proteobacteria | *Sphingomonas* sp. | HQ256855 | 71 |
| 55 | Marine | 33b09 | α-Proteobacteria | *Sphingomonas* sp. | HQ256873 | 71 |
| 72 | Marine | 47b03 | α-Proteobacteria | *Sphingomonas* sp. | | 71 |
| 87 | Marine | 60b33 | α-Proteobacteria | *Sphingomonas* sp. | KR922204 | 71 |
| 54 | Marine | 32b69 | γ-Proteobacteria | *Stenotrophomonas* sp. | HQ256865 | 71 |
| 55 | Marine | 33b01 | γ-Proteobacteria | *Stenotrophomonas* sp. | HQ256866 | 71 |
| 79 | Marine | 52b07 | γ-Proteobacteria | *Stenotrophomonas* sp. | | 71 |
| 87 | Marine | 60b14 | α-Proteobacteria | *Brevundimonas* sp. | KR922192 | 72 |
| 97 | Marine | 67b02 | Bacteroidetes | *Chryseobacterium* sp. | KR922253 | 72 |
| 47 | Marine | 28b09 | Basidiomycota | *Dioszegia hungarica* | HQ256891 | 72 |
| 84 | Marine | 57b18 | Bacteroidetes | *Flavobacterium* sp. | KR922118 | 72 |
| 54 | Marine | 32b04 | α-Proteobacteria | *Sphingomonas* sp. | HQ256841 | 72 |
| 54 | Marine | 32b39 | α-Proteobacteria | *Sphingomonas* sp. | JF706504 | 72 |
| 54 | Marine | 32b30 | γ-Proteobacteria | *Stenotrophomonas* sp. | JN176584 | 72 |
| 54 | Marine | 32b38 | γ-Proteobacteria | *Stenotrophomonas* sp. | JN176587 | 72 |
| 55 | Marine | 33b08 | γ-Proteobacteria | *Stenotrophomonas* sp. | HQ256870 | 72 |
| 23 | Polluted | 07b11 | Firmicutes | *Bacillus* sp. | | 31 |
| 53 | Polluted | 31b07 | Actinobacteria | *Rhodococcus* sp. | HQ256828 | 50 |
| 53 | Polluted | 31b08 | Actinobacteria | *Frigoribacterium* sp. | HQ256829 | 52 |
| 88 | Polluted | 61b06 | γ-Proteobacteria | *Stenotrophomonas* sp. | KR922206 | 54 |
| 53 | Polluted | 31b04 | γ-Proteobacteria | *Pseudomonas* sp. | HQ256826 | 56 |
| 53 | Polluted | 31b03 | Actinobacteria | *Kocuria palustris* | HQ256825 | 60 |
| 23 | Polluted | 07b06 | Firmicutes | *Paenibcillus* sp. | | 60 |
| 53 | Polluted | 31b02 | Actinobacteria | *Rhodococcus* sp. | HQ256824 | 62 |
| 88 | Polluted | 61b10 | Bacteroidetes | *Chryseobacterium* sp. | KR922207 | 66 |
| 53 | Polluted | 31b01 | Firmicutes | *Bacillus megaterium* | HQ256822 | 68 |
| 53 | Polluted | 31b09 | Actinobacteria | *Streptomyces* sp. | HQ256830 | 68 |
| 53 | Polluted | 31b10 | β-Proteobacteria | *Massilia* sp. | HQ256823 | 71 |

**References**

Deguillaume, L., Charbouillot, T., Joly, M., Vaïtilingom, M., Parazols, M., Marinoni, A., Amato, P., Delort, A.-M., Vinatier, V., Flossmann, A., Chaumerliac, N., Pichon, J. M., Houdier, S., Laj, P., Sellegri, K., Colomb, A., Brigante, M. and Mailhot, G.: Classification of clouds sampled at the puy de Dôme (France) based on 10 yr of monitoring of their physicochemical properties, Atmospheric Chem. Phys., 14(3), 1485–1506, doi:10.5194/acp-14-1485-2014, 2014.

[Figure]

**Figure 2.** Time profile of surface tension measurements in a non-logarithmic form.

[Figure]

**Figure 4 (a).** Surface tension (σ) distribution of the 480 strains tested for biosurfactant production according to the physicochemical characteristics of cloud waters (marine, highly marine, continental and polluted). Highlighted in blue, the number of tested strains. Box and whisker plots are shown with the minimal (red) and maximal (green) surface tensions. The orange boxes represent 25th and 75th percentiles (lower and upper quartiles) of the measure **(b).** Phyla distribution according to the physicochemical characteristics of cloud waters (marine, highly marine, continental and polluted).

**Manuscript acp-2016-447**
Screening of cloud microorganisms isolated at the puy de Dôme (France) station for the production of biosurfactants
P. Renard, I. Canet, M. Sancelme, N. Wirgot, L. Deguillaume, and A.-M. Delort

**Answer to Referee #1**
First we would like to thank to the reviewers for their work and interest in our work. We have taken into account their comments to improve the manuscript and answered point by point to their questions. Changes in the manuscript are underlined in yellow.

**Anonymous Referee #1**
The authors report equilibrium surface tension values for every microorganism in their study. However, it is unclear how they determine when equilibrium has been reached. All of the surface tension time profiles in Figure 2 appear to be decreasing when the measurements were stopped. That is, the reported equilibrium surface tension values are the minimum values for the time profiles given, but may not be if the time profile was extended. In section 2.3, the authors state a 30-minute maximum for surface tension measurements but give no justification for this time frame.

**Author's response**
You are absolutely right, the minimum of the equilibrium region ($\sigma_{eq}$), is difficult to determine experimentally (small variations of surface tension over long timescales) (Nozière et al., 2014). The surface tension decreases asymptotically and the logarithmic scale of the figure 2 is probably misleading, it looks clearer when presented in a non-logarithmic form (see enclosed Figure 2. Time profile of surface tension measurements in a non-logarithmic form.). The overall equilibration time, $t_{eq}$, is of the order of $2 \times t_m$ (time of meso-equilibrium) (Nozière et al., 2014). After this period, surface tension decreases marginally. According to the measurements we did on longer period (few hours), our overestimation is comprised between 0.1 and 0.2 mN m$^{-1}$. That is why, above 30 mN m$^{-1}$, we only give nominal surface tension (Table S2 in the supplementary). Below 30 mN m$^{-1}$ and above the CMC, surface tension decreases quickly and it is easier to be more accurate.
Finally, we decided to keep our original Figure 2 presented in the logarithmic mode as it is the usual way in publications and allows to show the initial surface tension ($\sigma_0$), the surface tensions in the meso-equilibrium ($\sigma_m$) and in the equilibrium ($\sigma_{eq}$) phase (See Noziere et al 2014 for instance).

**Anonymous Referee #1**
In Figure 2, the authors show the surface tension time profile for the R2A broth, which is the medium used for all cultures of their isolated microorganisms. However, this may not be a good baseline because incubation period lasts between six to ten days. We accept that the microorganisms are altering the composition of the broth by producing biosurfactants, but they are consuming nutrients in the broth as well. It remains to address how the removal of nutrients would impact the surface tension of the crude extracts.

**Author's response**
We agree that broth, i.e. carbon sources, influences biosurfactant production. This is true for industrial production of biosurfactants performed in aqueous media with a rich carbon source feedstock, such as carbohydrates, hydrocarbons, fats, and oils. Such enriched broths increase the production. However in our case R2A broth is very poor so when the microorganisms consume the carbon sources, it does not make a great change. We confirmed this point thanks to the following experiments: We did purification of few biosurfactants from microbial R2A cultures, and then we measured the surface tension of these pure compounds both in water and in R2A medium, the differences were marginal (confidential, to be published).

**Author's changes**
We modified the text as follows:

P 9 line 9: In this collection, we observed 34 strains (7%) that reduce the surface tension of the R2A broth below 30 mN m$^{-1}$.

**Anonymous Referee #1**
Furthermore, the authors present a large amount of data regarding the surface tension of crude extracts but do not make a connection to the surface tension of cloud water, which is arguably the basis for this work. Since the authors have already collected the cloud water in order to isolate the microorganisms, it would be useful to also report surface tension values for the cloud water samples as well extend the crude extract results to cloud water.

**Author's response**
We share your viewpoint; the surface tension of cloud water could have been relevant. Unfortunately, the cloud samplings have been performed before the acquisition of the tensiometer. However, according to the Köhler theory, the surface tension, as well as, the saturation vapor pressure and the CCN diameter, drive the activation of particles into cloud droplets. The activation occurs when the radius of the cloud droplet is minimal (few nm, i.e., wet aerosol) and the concentration of organic compounds, such as biosurfactant, is maximal (Nozière et al., 2014). The effect of surface tension is maximal during the activation. In cloud droplet (few μm), organic compounds are diluted, and biosurfactants are likely under the CMC. Nevertheless, measuring surface tension in concentrated cloud water could be a complementary work, especially since we observed in 300 fold-concentrated rain, a strong decrease of the surface tension (30 mN m$^{-1}$) (unpublished data).
Here, we demonstrate that bacteria sampled in clouds are able to produce biosurfactants under lab conditions. We are currently isolating and characterizing these biosurfactants. We have identified 11 different structures by mass spectrometry. In the future we want to collect cloud and rain samples and also aerosols and look for these structures in these atmospheric samples (this is what is proposed in the conclusion). This is a long term research plan.

**Author's changes**
In order to emphasize this point, we have modified the following sentences:

P 13 lines 4: "In conclusion, the results of the present study showed that the microbial strains isolated from cloud waters produce strong biosurfactants under laboratory conditions. The major and most active producers belong to the *Pseudomonas* genus, which is prevalent in cloud water and typically originates from the phyllosphere. Although the presence of surfactants has been shown on aerosols (Nozière et al., 2014), it has not yet been demonstrated in clouds, and the structure of these compounds has not been established. The biosurfactants overproduced by the best producers in the present study will be isolated to analyze their chemical structure. In parallel, the biosurfactants from cloud aerosols and rain samples will also be extracted, and their structural fingerprints will be analyzed and compared with the signatures of microbial surfactants isolated from clouds."

**Anonymous Referee #1**
Finally, the statistical analysis section did not seem to add much to the paper. The main takeaway was that α-Proteobacteria are efficient biosurfactant producers, which reinforces conclusions from section 3.2. However, the entire analysis seems unsubstantiated. The distinction between air mass origins seems arbitrary. The distinction between chemical compositions is more logical, but the conclusions for that analysis are weaker.

**Author's response**
Our statistics are based on 480 strains but these strains are grouped into 39 cloud events, thus partially dependent. This sampling was spread over 10 years, and represented with related analyzes, a considerable work and a more than correct observation of Puy de Dôme clouds.
However it is still difficult to make statistics on samples with such intra- and inter-sample variations. For example, in marine clouds, we identified only one strain in few events (e.g., event 29) compared to the 62 strains in the event 54 (see Table S1 in supplementary). This makes our Mann-Whitney and Kruskal-Wallis tests a bit weak.
We could use mixed model. Nevertheless, you are right, these statistics would not add much new, i.e., the correlation of *Pseudomonas* / surface tension. We therefore concluded it would be better to be limited to a high-quality observation.

In conclusion we decided to keep the paragraph "Impact of the origin and chemical composition of clouds on biosurfactant production" to give some general tendency. The obtained results are interesting as they suggest a link between the vegetation origin and the biosurfactant production. This should be studied in more details in the future.

**Author's changes**
In the abstract, we replaced:
Statistical analyses showed some positive correlations between the origin of air masses and chemical composition of cloud waters with the presence of biosurfactant-producing microorganisms, suggesting a "biogeography" of this production.
by:
We observed some correlations between the chemical composition of cloud water and the presence of biosurfactant-producing microorganisms, suggesting the "biogeography" of this production.

Page 4 line 5: we replaced:
"In order to evaluate the potential correlation between the origin of air masses and composition of cloud waters and the presence of biosurfactant-producing microorganisms, statistical analyses are performed."
by:
"We observed a potential correlation between the composition of cloud waters and the presence of biosurfactant-producing microorganisms."

P5 line 14: This text has been deleted:
**2.4 Statistical analyses**
Herein, we investigate the differences, in terms of impact on the non-normally distributed surface tension, due to the origin of air mass and the chemical composition of clouds using the PAST software version 3.09 (Hammer et al., 2001).
Using a non-parametric method, the Kruskal-Wallis one-way analysis of variance (Siegel, 1956), we compare the distributions of surface tensions between 4 air mass origin sectors: west (W), north-west/north (NW/N), north-east (NE) and south-west/south (SW/S) and between 4 chemical composition groups (Marine, Highly marine, Continental and Polluted). P-value < 0.05 is considered statistically significant.
Mann-Whitney test (Mann and Whitney, 1947), which is a measure of how different two populations are, allows specifying which group dominates, with two-by-two comparison.

Page 10 line 1: we totally rewrote the section 3.3 and replaced by:

**3.3 Potential impact of the chemical composition of the clouds on biosurfactant production**

In the present study, the screened microbial strains were isolated from 39 cloud events presenting different profiles. Information on the cloud chemical composition and the physicochemical parameters measured at the puy de Dôme station and described in (Deguillaume et al., 2014) is provided on the website of the Observatory of Earth Physics in Clermont-Ferrand (http://wwwobs.univ-bpclermont.fr/SO/beam/data.php). The main parameters, including pH, $SO_4^{2-}$, $NO_3^-$, $Cl^-$, acetate, formate, oxalate, succinate, malonate, $Na^+$, $NH_4^+$, $Mg^{2+}$, $K^+$, and $Ca^{2+}$, are summarized in the Supplemental materials (Table S1). These physico-chemical parameters were used for the ACP analysis as described in Deguillaume et al. (2014). The ACP generated 4 different types of clouds, classified as "highly marine", "marine", "continental" and "polluted". Typically, the more "polluted" clouds have a lower pH and higher concentrations of $NH_4^+$, $NO_3^-$, and $SO_4^{2-}$. The more "marine" clouds have a higher concentration of NaCl. The 39 cloud events were divided into 2 "highly marine", 26 "marine", 8 "continental" and 3 "polluted" clouds (Table S1).

[Figure]

**Figure 4 (a).** Surface tension (σ) distribution of the 480 strains examined for biosurfactant production according to the physicochemical characteristics of cloud waters (marine, highly marine, continental and polluted). Highlighted in blue, the number of tested strains. Box and whisker plots are shown with the minimal (red) and maximal (green) surface tensions. The orange boxes represent the 25th and 75th percentiles of the measurements **(b).** Phyla distribution according to the physicochemical characteristics of the cloud waters (marine, highly marine, continental and polluted).

Figure 4a shows the distribution of the surface tensions values (σ) measured from the 480 strains examined for biosurfactant production according to the cloud water chemical composition (marine, highly marine, continental or polluted). A comparison of the distribution of the phyla of the strains in the same cloud events is presented in Figure 4b. The samples from marine clouds constitute the majority of this collection (323/480 strains). We observed a difference between the surface tension values from continental and highly marine strains (medians: 56 and 61 mN m$^{-1}$, respectively). Highly marine clouds are characterized by the highest minimal surface tension (45 mN m$^{-1}$, Figure 4a), consistent with the almost complete absence of ϒ-Proteobacteria, which are the most efficient biosurfactant-producing microorganisms (σ ≤ 45 mN m$^{-1}$) (1/57 isolates, see Figure 4b). These observations were based on 39 cloud events with 480 different strains, representing, to our knowledge, the largest cloud sample data set studied; this data set is representative of cloud sampling over more than 10 years at the puy de Dôme station. Although it remains difficult to generate statistics on samples with such intra- and inter-sample variations, these results provide a general tendency that could be reinforced and confirmed with more data in the future.

Figure 4, Figure 5a, Figure 6a, Table S3 and Figure S1 have been deleted.
We have kept only Figure 5b and Figure 6b which are now Figure 4(a) and (b) in the revised manuscript, note that the presentation of the data has been modified as suggested by referee 2.

**Anonymous Referee #1**
The grammatical errors are too numerous to list individually. This paper would greatly benefit from editing by a native English speaker.
**Authors:** The manuscript has been proof read by ACS services

Page 5, line 2. Might be helpful to keep units consistent with Page 4, line 33. Either g (preferred) or rpm.
**Authors:** We put these values: 10,480 g / 3 min

Page 5, line 11. Change section number from Roman to Arabic numerals.
**Authors:** done

Page 9, lines 9-10. This is a misrepresentation because the biosurfactants are reducing the surface tension of the R2A broth, not pure water.
**Authors:** Changed

Page 9, line 11. I think you mean surface tension values between 30 and 45 mN m$^{-1}$ not up to 45 mN m-1.
**Authors:** Yes, we agree, we have changed it to "between 30 and 45 mN m$^{-1}$"

Page 9, line 18. Third and fourth is clearer than third and last.
**Authors:** Changed

Page 11, lines 12-14. There is not a significant difference between all four sectors, just between NW/N and the others, according to your supplementary information.
**Authors**: Actually we have deleted all the data linked to the back-trajectories of the air masses (see answer to referee 2)

Page 14, lines 3-5. Citation for this sentence?
**Authors**:  we have added  "(Joly et al., 2015 and references hereinafter)".

**Manuscript acp-2016-447**

Screening of cloud microorganisms isolated at the puy de Dôme (France) station for the production of biosurfactants

P. Renard, I. Canet, M. Sancelme, N. Wirgot, L. Deguillaume, and A.-M. Delort

**Answer to Referee #2**

First we would like to thank to the reviewers for their work and interest in our work. We have taken into account their comments to improve the manuscript and answered point by point to their questions. Changes in the manuscript are underlined in yellow.

**Anonymous Referee #2**

English grammar: This paper must be completely proofread by a native English speaker before publication should be considered.

**Author's changes**

The manuscript has been proof read by ACS services

**Anonymous Referee #2**

-Clouds: the largest hole in this manuscript is the lack of cloud-water analysis. The authors acceptably demonstrate that biosurfactant producing bacteria exist in their cloud water samples but fail to demonstrate if the bacteria actually have a measurable effect on their collected cloud droplets. Even if the authors are unable to measure surface tension depression in their cloud samples, this should still be noted and contextualized in the manuscript. The paragraph in the discussion section, starting P14-L33, would greatly benefit from this analysis.

**Author's response**

We deeply agree with this comment. To demonstrate if the bacteria have a measurable effect on the formation of cloud droplets is the critical issue, and the point of further studies.

Nozière et al. (2014) observed strong decreases of surface tension on aerosols: "The first results have shown that these fractions are much more surface-active than expected ($\sigma$: 30 mN m$^{-1}$) and display properties similar to those of biosurfactants such as surfactin or rhamnolipids".

Here, we demonstrate that bacteria sampled in clouds are able to produce biosurfactants under lab conditions. We are currently isolating and characterizing these biosurfactants. We have identified 11 different structures by mass spectrometry. In the future we want to collect atmospheric water samples (rain and cloud) and also aerosols and look for these structures in these atmospheric samples (this is what is proposed in the conclusion). This is a long term research plan.

We would like to add that, although we did not measure surface tension in cloud waters, we recently measured it in concentrated rain (x300) and found a value of 30 mN/m$^{-1}$.

**Author's changes**

In order to emphasize this point, we have modified the following sentences:

P 11 Line 7: "In the present study, we showed that under laboratory conditions, the most efficient biosurfactant-producing microorganisms ($\sigma$ < 45 mN m$^{-1}$) belonged to a limited number of bacterial

genera (*Pseudomonas* and *Xanthomonas)* from the ϒ-Proteobacteria class (78%) and a yeast genus (*Udeniomyces*) from the Basidiomycota phylum (11%)."

P 13 lines 4: "In conclusion, the results of the present study showed that the microbial strains isolated from cloud waters produce strong biosurfactants under laboratory conditions. The major and most active producers belong to the *Pseudomonas* genus, which is prevalent in cloud water and typically originates from the phyllosphere. Although the presence of surfactants has been shown on aerosols (Nozière et al., 2014), it has not yet been demonstrated in clouds, and the structure of these compounds has not been established. The biosurfactants overproduced by the best producers in the present study will be isolated to analyze their chemical structure. In parallel, the biosurfactants from cloud aerosols and rain samples will also be extracted, and their structural fingerprints will be analyzed and compared with the signatures of microbial surfactants isolated from clouds."

**Anonymous Referee #2**
Arbitrary choices: While not debilitating to the paper itself, two arbitrary divisions are made in this manuscript: 1) the surface tension division of Fig.3 … For the surface tension divisions, the authors cite Baudel et al. (2012) and Ekstrom et al. (2010) saying that their divisions are chosen in a similar way. However, neither Baudel nor Ekstrom divide their data in the same way that the authors here are trying to do. It would be more correct to say that the authors are choosing bins for their samples that match 1-2 bins of previously published works. The other bins are completely arbitrary and the authors do not provide any reasons for why they chose >55, 45-55, 30-45, and <30 mN/m. Some reasoning behind these bins needs to be present in the manuscript.

**Author's response**
You are right, Ekström et al. (2010) and Baduel et al. (2012) have not, strictly speaking, defined categories. Our categories rather correspond to the values observed by them. As cited thereafter the value of 30 mN m$^{-1}$ refers to strong biosurfactants, the value of 45 mN m$^{-1}$ to HULIS. We agree that the value of 55 mN m$^{-1}$ is not so well defined and is more arbitrary. Note that our comments in the text greatly refer to values < 45 mN m$^{-1}$ .
Furthermore, we agree, to categorize quantitative variables is always somewhat arbitrary but it helps to present the results.

According to Ekström et al. (2010), 30 mN m$^{-1}$ is the distinct signature of microbial surfactants: *"The very low surface tension values obtained with the aerosol samples were attributed to the presence of biosurfactants, because these compounds are the only natural substances able to have such strong effects on the surface tension"*
*"Comparing the curves for the standard compounds with those obtained with the aerosol extracts on Fig. 3 (curves with open circles) clearly show that the latter have the distinct signatures of microbial surfactants: a surface tension below 30 mN m$^{-1}$ at high concentrations, and a sharp transition characteristic of micelle-forming surfactants".*

Baduel et al. (2012) also observed strong decrease of surface tension on their atmospheric aerosol samples between 30 and 45 mN m$^{-1}$): *"The minimum surface tension obtained from the summer samples was systematically lower (30 mN m$^{-1}$) than that of the winter samples (35-45 mN m$^{-1}$)."*

According to Ekström et al. (2010), humic-like substances (HULIS) would only lower the surface tension to 45 mN m$^{-1}$: *"This implies that only a few tens of µM of biosurfactants would lower the surface tension of water to about 30 mN m$^{-1}$. By contrast, 20mM (~ 20 g L$^{-1}$) of HULIS would only lower the surface tension to 45 mN m$^{-1}$ (Taraniuk et al., 2007), and 10M of malonic acid would lower it to 50 mN/m."*.

55 mN m$^{-1}$ is probably the most arbitrary limit; it approximates the first surface tension values measured on aerosols filter samples: *"So far, the only way to perform such measurements is with aerosol filter samples, which are extracted, and the surface tension of the extracts measured with a tensiometer. The first studies using these methods reported surface tensions between 52 mN m$^{-1}$ (Mircea et al., 2005) and 60 mN m$^{-1}$ (Capel et al., 1990; Facchini et al., 1999, 2000; Hitzenberger et al., 2002; Decesari et al., 2005; Mircea et al., 2005). The small amounts of material on the filters made these methods challenging. An additional drawback was that the extraction, usually in water, was not specific to surfactants, leading to mixtures where the contribution of the surfactants was underestimated."* (Baduel et al., 2012).

**Author s' changes**
According to the reviewer remarks we have modified the text as follows:
P 9 line 7: we deleted this sentence: "These 4 categories are chosen in a similar way to Baduel et al. (2012) and Ekström et al. (2010)."

P 9 line 8: we modified the text as follow:
"The first category (σ ≤ 30 mN m$^{-1}$) is rare among man-made surfactants and is typical of surfactants of biological origin (Christofi and Ivshina, 2002). In this collection, we observed 34 strains (7%) that reduce the surface tension of the R2A broth below 30 mN m$^{-1}$. These strains exclusively belonged to the genera *Pseudomonas* and *Xanthomonas* (Υ-Proteobacteria, Fig. 1b). The second category corresponded to surface tension values between 30 and 45 mN m$^{-1}$. The 55 mN m$^{-1}$ limit is often considered the threshold in terms of the surface tension decrease originating from HULIS (humic like substances) (Kiss et al., 2005; Taraniuk et al., 2007). We observed only 30 strains (6%) in this second category. In summary, from the first two categories (σ ≤ 45 mN m$^{-1}$), although new phyla were observed in the second category, the phylum distribution of the most efficient biosurfactant-producing microorganisms remains largely dominated by Υ-Proteobacteria (78% of all strains) and more moderately by Basidiomycota (11%) (Fig. 3). Notably, the two other major taxa of all studied strains, Actinobacteria and α-Proteobacteria, almost completely disappear in these categories. The third and fourth categories (45 < σ ≤ 55 and σ > 55 mN m$^{-1}$) represented 28 and 59% of the collection, respectively. The 55 mN m$^{-1}$ limit is relatively arbitrary but approximates the first surface tension values measured on the aerosol filter samples (Baduel et al., 2012; Capel et al., 1990; Decesari et al., 2005; Facchini et al., 1999, 2000; Hitzenberger et al., 2002; Mircea et al., 2005)."

**Anonymous Referee #2**
-Arbitrary choices: While not debilitating to the paper itself, two arbitrary divisions are made in this manuscript: […] 2) the source region division of Fig. 4 For the source region divisions, the authors present no reasoning for dividing the air masses into marine/highly marine/etc. From Table S1, it is impossible to tell why marine and highly marine are split. Fig. S1 suggests the split is because of time over open ocean but the authors need to be explicit here.
I would also argue that Fig. 4 adds nothing to the manuscript and should be replaced with the HYSPLIT trajectories and some additional meteorological statistics (e.g. wind direction histogram for the sampling site for both cloud-sampling days and non-sampling days). There is also no reason to not show the air mass height results from HYSPLIT.

**Author's response**

We understand from the reviewer's remarks that our text was not clear and did not give enough details to be understandable.

The different categories as described in Figure 4 are based on the paper of Deguillaume et al. (2014). In this paper the air masses are defined according two different types of criteria:

1) **Their back trajectories** : They are calculated using the HYSPLIT (Hybrid Single-Particle Lagrangian Integrated Trajectory) model with the GDAS1 meteorological data archive and default settings (Draxler and Rolph, 2010). Considering 10 years of monitoring at the puy de Dôme station, Deguillaume et al (2014) divided France in four sectors to classify these back trajectories crossing France (West, North East, NorthWest/North, SouthWest/West).

2) **The chemical content of cloud water:** The physicochemical parameters presented in **Table S1** (pH, $SO_4^{2-}$, $NO_3^-$, $Cl^-$, Acetate, Formate, Oxalate, Succinate, Malonate, $Na^+$, $NH_4^+$, $Mg^{2+}$, $K^+$,$Ca^{2+}$) are used to make an ACP analysis as described in Deguillaume et al. (2014). This ACP gives 4 different groups which have been named "highly marine", "marine", "continental" and "polluted". Typically, the more "polluted" are the clouds, the lower is the pH and the higher are the concentrations of $NH_4^+$, $NO_3^-$, $SO_4^{2-}$. The more "marine" are the clouds, the higher is the concentration of NaCl.

In our opinion the second type of categories based of chemical measurements is more accurate as it reflects the whole history of the clouds integrating their eventual complex trajectories.

**Author's changes**

Therefore, in the revised manuscript we deleted all the data linked to the back trajectories and only kept the "highly marine", "marine", "continental" and "polluted" categories. This includes deletion of "W, NE, NW, NW/N and SW/W" in Tables S1 and S2, of Table S3, of Figure S1, of Figure 4, of Figure 5(a) and Figure 6 (a).

Figures 5b and Figure 6b are now Figures 4(a) and (b) in the revised manuscript.

P 10, line 11, this text was deleted: "Cloud events are first classified according to the air mass origin - *i.e.*, west (W), north-west/north (NW/N), north-east (NE) and south-west/south (SW/S) - determined from backward trajectories (see Fig. 4 and Fig. S1 in the supplement). Second, cloud events are classified according to the physicochemical characteristics of cloud waters (Marine, Highly marine, Continental and Polluted, see Fig S1 and Table S1 in the supplement) as described by Deguillaume et al. (2014).

Figure 4 shows that 18 events come from W consisting of 2 Highly marine, 14 Marine and 2 Continental cloud events, 13 events from NW/N with 10 marine, 2 continental and 1 polluted cloud events, 3 events from NE with 2 polluted cloud events and the other continental and, from the SW/S, 5 events of which 2 marine and 3 continental cloud events."

P 11 line 10, this text was deleted: Figure 5a shows the distribution of surface tensions values (σ) measured from the 480 strains tested for biosurfactant production, according to the air mass origins (4 sectors: W, NW/N, NE and SW/S). Samples from west sector constitute the great majority of our collection (318/480 strains). From statistical analysis, we observe a significant difference (Kruskal-Wallis p-value: 0.0049 << 0.05) in the distribution of biosurfactant-producing microorganisms between the NW/N sectors and the others (W, NE and SW/S). The Mann-Whitney test (see details in the Supplement, Table S3) allows us to attribute this difference to the NW/N sector with a surface tension median (53 mN m$^{-1}$) significantly lower (Mann-Whitney p-values < 0.05) than the other three sectors (medians: 59, 60 and 61 mN m$^{-1}$, for W, NE and SW/S sectors, respectively). This difference cannot be completely attributed to the differences in the phyla distribution within the different air masses (Fig. 6). Indeed, as shown before, the most efficient biosurfactant-producing microorganisms belong to γ-Proteobacteria class, which represents 23% of all strains, but its distribution regarding the air mass origin sectors remains unselective (26, 28, 3 and 14% for W, NW/N, NE and SW/S sectors, respectively). The difference in the distribution of biosurfactantproducing microorganisms between the four sectors is rather due to the proportion of the most efficient biosurfactant-producing microorganisms ($\sigma \leq 45$ mN m$^{-1}$) amongst the ϒ-Proteobacteria class (Fig. 6). In the NW/N sector, most efficient biosurfactant-producing microorganisms account for 68% of ϒ-Proteobacteria (13/19 isolates), against 40% in the other three sectors (37/93 isolates). No such difference amongst the ϒ-Proteobacteria class is observed in the chemical composition groups.

P 10 line 6, We have added these sentences in the modified manuscript:

"The main parameters, including pH, SO$_4^{2-}$, NO$_3^-$, Cl$^-$, acetate, formate, oxalate, succinate, malonate, Na$^+$, NH$_4^+$, Mg$^{2+}$, K$^+$, and Ca$^{2+}$, are summarized in the Supplemental materials (Table S1). These physico-chemical parameters were used for the ACP analysis as described in Deguillaume et al. (2014). The ACP generated 4 different types of clouds, classified as "highly marine", "marine", "continental" and "polluted". Typically, the more "polluted" clouds have a lower pH and higher concentrations of NH$_4^+$, NO$_3^-$, and SO$_4^{2-}$. The more "marine" clouds have a higher concentration of NaCl. The 39 cloud events were divided into 2 "highly marine", 26 "marine", 8 "continental" and 3 "polluted" clouds (Table S1)."

**Anonymous Referee #2**
Furthermore, the analysis starting on page 11, which attributes statistical difference between the air mass divisions (one of which has 3 samples!), seems weak. I would suggest that there are stronger ways to segregate the air masses given the Table S1. I believe the whole analysis should be rerun using the chemical speciation data.
Shorter Comments:
-Mann-Whitney and KW ANOVA – can the authors comment on how appropriate it is to assume statistical independence for air masses that, though they are measured at the sampling site as from 2 different directions, shared the same path as few as 5 days prior to sampling?
-Figure 5 needs to be replaced with a proper box and whiskers plot or some other way to judge what the real spread in the data looks like (possibly note the std. dev.?)

**Author's response**
As explained above, Deguillaume et al (2014) classifies clouds according to two different criteria: back trajectories and physico-chemical parameters. In our opinion the second type of categories based of chemical measurements is more accurate as it reflects the whole history of the clouds integrating their eventual complex trajectories. Therefore, in the revised manuscript we have deleted all the data and text linked to the back trajectories and only kept the "highly marine", "marine", "continental" and "polluted" categories.
Our statistical analyses are based on 39 cloud events with 480 different strains. This represents, to our knowledge, the largest data set on cloud samples ever studied; it is representative of cloud sampling over more than 10 years at the puy de Dôme station. These cloud events when classified according to ACP analysis (Deguillaume et al. 2014) are independent and can be compared. The only problem is that it is still difficult to make statistics on samples with such intra- and inter-sample variations (not because there is only 3 samples for example). For example, in marine clouds, we identified only one strain in few events (e.g., event 29) compared to the 62 strains in the event 54 (see Table S1 in supplementary).This makes our Mann-Whitney and Kruskal-Wallis tests a bit weak.
We have therefore concluded it would be better to be limited to a high-quality observation rather than making questionable statistics, which would not adding much new, i.e., the correlation of *Pseudomonas* / surface tension.
 In conclusion we decided to keep the paragraph "Impact of the origin and chemical composition of clouds on biosurfactant production" to give some general tendency. The obtained results are interesting as they suggest a link between the vegetation origin and the biosurfactant production. This should be studied in more details in the future.

**Author's changes**

In the abstract, we replaced:
Statistical analyses showed some positive correlations between the origin of air masses and chemical composition of cloud waters with the presence of biosurfactant-producing microorganisms, suggesting a "biogeography" of this production.
by:
We observed some correlations between the chemical composition of cloud water and the presence of biosurfactant-producing microorganisms, suggesting the "biogeography" of this production.

Page 4 line 5: we replaced:
"In order to evaluate the potential correlation between the origin of air masses and composition of cloud waters and the presence of biosurfactant-producing microorganisms, statistical analyses are performed."
by:
"We observed a potential correlation between the composition of cloud waters and the presence of biosurfactant-producing microorganisms."

P5 line 14: This text has been deleted:
**2.4 Statistical analyses**
Herein, we investigate the differences, in terms of impact on the non-normally distributed surface tension, due to the origin of air mass and the chemical composition of clouds using the PAST software version 3.09 (Hammer et al., 2001).
Using a non-parametric method, the Kruskal-Wallis one-way analysis of variance (Siegel, 1956), we compare the distributions of surface tensions between 4 air mass origin sectors: west (W), north-west/north (NW/N), north-east (NE) and south-west/south (SW/S) and between 4 chemical composition groups (Marine, Highly marine, Continental and Polluted). P-value < 0.05 is considered statistically significant.
Mann-Whitney test (Mann and Whitney, 1947), which is a measure of how different two populations are, allows specifying which group dominates, with two-by-two comparison.

Page 10 line 1: we totally rewrote the section 3.3 and replaced by:

**3.3 Potential impact of the chemical composition of the clouds on biosurfactant production**

In the present study, the screened microbial strains were isolated from 39 cloud events presenting different profiles. Information on the cloud chemical composition and the physicochemical parameters measured at the puy de Dôme station and described in (Deguillaume et al., 2014) is provided on the website of the Observatory of Earth Physics in Clermont-Ferrand (http://wwwobs.univ-bpclermont.fr/SO/beam/data.php). The main parameters, including pH, $SO_4^{2-}$, $NO_3^-$, $Cl^-$, acetate, formate, oxalate, succinate, malonate, $Na^+$, $NH_4^+$, $Mg^{2+}$, $K^+$, and $Ca^{2+}$, are summarized in the Supplemental materials (Table S1). These physico-chemical parameters were used for the ACP analysis as described in Deguillaume et al. (2014). The ACP generated 4 different types of clouds, classified as "highly marine", "marine", "continental" and "polluted". Typically, the more "polluted" clouds have a lower pH and higher concentrations of $NH_4^+$, $NO_3^-$, and $SO_4^{2-}$. The more "marine" clouds have a higher concentration of NaCl. The 39 cloud events were divided into 2 "highly marine", 26 "marine", 8 "continental" and 3 "polluted" clouds (Table S1).

[Figure]

**Figure 4 (a).** Surface tension (σ) distribution of the 480 strains examined for biosurfactant production according to the physicochemical characteristics of cloud waters (marine, highly marine, continental and polluted). Highlighted in blue, the number of tested strains. Box and whisker plots are shown with the minimal (red) and maximal (green) surface tensions. The orange boxes represent the 25th and 75th percentiles of the measurements **(b).** Phyla distribution according to the physicochemical characteristics of the cloud waters (marine, highly marine, continental and polluted).

Figure 4a shows the distribution of the surface tensions values (σ) measured from the 480 strains examined for biosurfactant production according to the cloud water chemical composition (marine, highly marine, continental or polluted). A comparison of the distribution of the phyla of the strains in the same cloud events is presented in Figure 4b. The samples from marine clouds constitute the majority of this collection (323/480 strains). We observed a difference between the surface tension values from continental and highly marine strains (medians: 56 and 61 mN m$^{-1}$, respectively). Highly marine clouds are characterized by the highest minimal surface tension (45 mN m$^{-1}$, Figure 4a), consistent with the almost complete absence of Ɣ-Proteobacteria, which are the most efficient biosurfactant-producing microorganisms (σ ≤ 45 mN m$^{-1}$) (1/57 isolates, see Figure 4b). These observations were based on 39 cloud events with 480 different strains, representing, to our knowledge, the largest cloud sample data set studied; this data set is representative of cloud sampling over more than 10 years at the puy de Dôme station. Although it remains difficult to generate statistics on samples with such intra- and inter-sample variations, these results provide a general tendency that could be reinforced and confirmed with more data in the future.

Figure 4, Figure 5a, Figure 6a, Table S3 and Figure S1 have been deleted.
We have kept only Figure 5b and Figure 6b which are now Figure 4(a) and (b) in the revised manuscript, note that the presentation of the data has been modified as suggested by the referee (whiskers plots).

**Anonymous Referee #2**
Paragraph starting P15, L26 – I am generally a proponent of contextualizing your work but I am not convinced this paragraph adds anything to the manuscript. It should either be removed or condensed and moved to the introduction.

**Author's response**
We agree with the referee that this paragraph is a bit out of scope of this paper, therefore it has been deleted.

**Author's changes**
Page 15 line 26, this paragraph has been deleted:
To our knowledge, research on the potential impact of biosurfactants on human health due to their presence in atmospheric waters remains marginal, compared to terrestrial or aquatic ecosystem studies (Olkowska et al., 2014). Aerosols are now well-known to represent a major concern for the populations as shown by epidemiology studies (Bernstein et al., 2004; Dominici et al., 2014; Pope, 2000). The toxicological impact is inversely proportional to the particle size (Nel, 2005; Novák et al., 2014). For example, fine particles ($PM_{1-2.5}$) reach lung alveoli and ultrafine particles ($PM_{0.1}$), once in the lung, would readily pass into the bloodstream and cause a direct insult to the cardiovascular system and other organs (Moshammer and Neuberger, 2003; Polichetti et al., 2009). The composition of the aerosols is also of major importance and should not be underestimated (Brimblecombe and Latif, 2004; Škarek et al., 2007). Regarding more precisely surface-active organic aerosols, few reports are devoted to synthetic molecules. Thus, Poulsen et al. (2000) suggested that molecules with surfactant properties could interfere in the immunological pathways, which could explain the increase of allergic diseases in industrialized societies. At high concentration in the atmosphere, surfactants can lead to asthma, can disrupt the stability of human respiratory systems, as well as can cause dry eyes allergies (Ahmad et al., 2009; Xinxin et al., 2016). Rhamnolipids inhibit ciliary function and produce damage to the human bronchial epithelium (Abdel-Mawgoud et al., 2011). Surfactants, by lowering the surface tension of tear films of the eyes could also be at the origin of dry eyes sensation (Vejrup and Wolkoff, 2002). Hence, biosurfactants present on aerosols could have a double impact on human cells, first because they could destroy cell membranes of the host, second because they could concentrate and dissolve toxic pollutants and help their penetration within the host cells. Further studies are needed to better evaluate the impact of surfactant on human health.

**Anonymous Referee #2**
Minor comments:
- S2.2, P4, L23-25 : Given that most of the people reading ACP are not biologists, a one sentence explanation on why those three specific agars were chosen should be added to the text. Also, TSA should be defined.
-P5, L11: Should read "3.2" not "III.2"

**Author's changes**
- Page 4 line 18, the text has been modified as follows:
"Triplicate volumes of 0.1 mL of cloud water were plated onto R2A agar growth medium (Reasoner and Geldreich, 1985; DIFCO™), and eventually onto R2A medium supplemented with NaCl 20 g $L^{-1}$ and King's B (King et al., 1954), Sabouraud (DIFCO™) and TSA (Trycase Soy Agar, DIFCO™) media. The plates were incubated at 17°C or 5°C under aerobic-dark conditions until the appearance of colonies (typically 6 days

at 17°C or 10 days at 5°C) (Vaïtilingom et al., 2012). R2A medium is a poor medium initially developed to isolate microorganisms from tap water and is well adapted to cloud samples, which are also poor. The addition of NaCl to R2A favors the selection of marine microorganisms; King's B medium is selective for *Pseudomonas* strains, while Sabouraud medium is selective for yeast strains."

- Page 5, Line 1,  Should read "3.2" not "III.2":
Done

---

## Author Response (AR2)

**acp-2016-447    Screening of cloud microorganisms isolated at the puy de Dôme (France) station for the production of biosurfactants**

P. Renard, I. Canet, M. Sancelme, N. Wirgot, L. Deguillaume, and A.-M. Delort

**Referee 1 #'s comments**

The authors have sufficiently addressed the comments of the reviewers and it is recommended that this work be published after some minor revisions:

Page 1, Line 26. The paragraph on human health has been removed based on Referee #2's comments, so the reference to human health in the abstract should also be removed.

Page 8, Line 12. I recommend replacing "a more or less strong" with something like "a noticeable" or "an appreciable" for clarity, though this more of a personal preference.

Page 9, Line 14. I think you mean the 45 mN m-1 limit, as mentioned in the author's response to Referee #2, not the 55 mN m-1 limit.

Page 12, Line 3. Since the statistical analysis based on air mass origins has been removed, this paragraph, especially the comment regarding the Northwest-North sector, is out of context. Please revise.

**Author's answer**

We thank the reviewer for his comments. We have corrected the revised manuscript accordingly

**Author's changes**

The changes are highlighted in yellow in the revised manuscript

P1: "Moreover, the potential impact of the production of biosurfactants by cloud microorganisms on atmospheric processes is discussed."

P 8: "As expected, a noticeable reduction of the surface tension values was observed when the microorganisms produced biosurfactants in the culture medium."

P9: "The 45 mN m$^{-1}$ limit is often considered the threshold in terms of the surface tension decrease originating from HULIS…"

P 12: "For example, French regions characterized by uniform monocultures (Ministère de l'Agriculture, de l'Agroalimentaire et de la Forêt, 2014) could be extremely favorable to plant pathogens, such as *Pseudomonas syringae* (McDonald and Linde, 2002)."